# You Do Not Fully Utilize Transformer's Representation Capacity

## Abstract

In contrast to RNNs, which compress their history into a single hidden state, Transformers can attend to all past tokens directly. However, standard Transformers rely solely on the hidden state from the previous layer to represent the entire context. We show that this design choice induces representation collapse and degrades performance. To address this issue, we introduce *Layer-Integrated Memory* (LIMe), a lightweight extension that leverages existing key–value buffers and learns per-head, per-layer routing weights to integrate representations from all previous layers with negligible overhead. Through extensive experiments—including language modeling, synthetic reasoning benchmarks, and very deep architectures—LIMe consistently achieves faster convergence, lower perplexity per FLOP, and substantial accuracy improvements on synthetic tasks while preserving higher value–vector entropy and improved token separability. Finally, our analysis of the learned routing weights reveals systematic reuse of both local and long-distance features, demonstrating how LIMe mitigates collapse, unlocks richer representations without increasing hidden-state size, and points to promising directions for future research.

## 1 Introduction

Transformers (Vaswani et al., 2017) have become a central architecture in modern machine learning, powering state-of-the-art solutions in language modeling, computer vision, and beyond. Their ability to capture complex patterns arises from deeply stacked layers that refine contextual representations. However, despite their success, standard Transformer decoders maintain a single residual stream per layer, forcing the model to compress all previously learned features into the immediately preceding hidden state (Srivastava et al., 2015; He et al., 2015). This design choice can lead to *representation collapse*—a phenomenon in which different tokens or features become indistinguishable in deeper layers (Voita et al., 2019; Barbero et al., 2024; Arefin et al., 2024). The problem is particularly pronounced when learning from lengthy sequences, where subtle token distinctions risk being squeezed out by limited floating-point precision and finite hidden-state capacity.

In this paper, we propose *Layer-Integrated Memory* (**LIMe**), a lightweight extension to multi-head self-attention that enables each attention head to retrieve and integrate representations from all preceding layers—rather than relying solely on the most recent hidden state. LIMe accomplishes this by learning a per-layer, per-head routing mechanism that efficiently blends multi-layer Key–Value features, all while preserving the core Transformer structure and adding negligible overhead by reusing already allocated Key–Value buffers.

Our key contributions are:

- **Layer-Integrated Routing.** A trainable router that, for each head at every layer, dynamically weights and mixes buffered Key–Value representations from all earlier layers, without increasing hidden-state dimensions or memory footprint.

- **Strong Empirical Gains.** LIMe converges 15.3% (8.9% with GQA) faster in FLOPs and achieves 1.15% (0.91% with GQA) lower perplexity than 1B-parameter LLaMa-based (Grattafiori et al., 2024) transformer, yields up to +8% on ProsQA (Hao et al., 2024) and +30% on arithmetic reasoning benchmarks (Arefin et al., 2024; Feng et al., 2023). In deep

settings (32, 64, 128 layers), a 64-layer LIMe matches a 128-layer baseline, indicating superior scaling behavior.

- **Mitigating Collapse.** An empirical analysis showing that LIMe preserves higher Rényi entropy (Arefin et al., 2024) and better token separability (Voita et al., 2019) in value spaces, effectively alleviating representation collapse.

Together, these results confirm that by distributing representational burden across persistent Key–Value buffers and learning to route information across layers, LIMe substantially improves both optimization efficiency and representational capacity, especially in tasks requiring long-range or multi-step reasoning, opening the door of utilizing LIMe for cutting-edge area of latent-space reasoning.

## 2 RELATED WORK

Early works on training very deep networks highlighted the need for mechanisms to ease gradient flow and information propagation. Highway Networks introduce gated skip connections to regulate information flow across layers (Srivastava et al., 2015). Deep Residual Networks further simplify this by adding identity shortcuts, enabling networks to exceed a hundred layers without suffering from vanishing gradients (He et al., 2015). Transformers adopt a similar residual-plus-normalization design, which underpins their success in language and vision tasks (Vaswani et al., 2017; Grattafiori et al., 2024; Jiang et al., 2023; Qwen et al., 2024; DeepSeek-AI et al., 2024).

Although residual streams facilitate training, they still force each layer to compress all prior features into a single vector, which can lead to *representation collapse*—distinct inputs becoming indistinguishable in deeper layers. Tenney et al. (2019) found that BERT's deeper layers refine earlier predictions using higher-level context. Voita et al. (2019) empirically demonstrated that Transformers' top layers lose fine-grained token distinctions. Theoretically, Barbero et al. (2024) proved that decoder-only Transformers can exhibit arbitrarily close final-token representations for different inputs, a phenomenon akin to *over-squashing*. Building on this, Hahn & Rofin (2024) showed that the loss landscape of Transformers biases them toward low-sensitivity functions, exacerbating collapse. Recently, Arefin et al. (2024) introduced Seq-VCR, a variance–covariance regularizer that preserves intermediate representation diversity and significantly improves multi-step reasoning performance.

To mitigate collapse, several works have explored aggregating information across layers. Cross-Layer Retrospective Retrieving learns dynamic attention weights over prior layer outputs for each head (Fang et al., 2023). Hyper-Connections augment Transformers with multiple residual streams that interact via learned projections, preventing collapse at the cost of increased hidden-state size (Zhu et al., 2024). LAuReL (Learned Augmented Residual Layer) generalizes the residual stream by introducing learned augmentations of the skip and, in variants that aggregate previous activations, by accessing hidden states from earlier layers during inference (Menghani et al., 2025). DenseFormer proposes using a weighted average of the previous layers' outputs as the input to each subsequent layer (Pagliardini et al., 2024). Value Residual Learning (ResFormer / SVFormer) reuses the first layer's value vectors across depth to improve attention concentration and KV efficiency (Zhou et al., 2025). Although Mixture-of-Depths (Raposo et al., 2024) focuses on reducing FLOPs by skipping token computations layer-wise, its dynamic routing approach resonates with our per-head, per-layer routing mechanism; unlike MoD, LIMe retains full dense computation while enriching representational capacity through routing over pre-allocated key–value buffers. Different architectures based on usage of previous representations were proposed in (Huang et al., 2018; Bapna et al., 2018; Wu et al., 2023). Despite these advances, most methods require substantial architectural changes or extra memory. Our method, Layer-Integrated Memory (LIMe), instead *reuses* existing key–value buffers and learns per-head, per-layer routing to mix multi-layer representations with negligible memory and speed overhead (see Appendix I).

## 3 PRELIMINARIES

**Notation.** Let $t$ denote the sequence length (temporal dimension), $d$ the model dimension, $H$ the number of attention heads, $d_{\text{head}} = d/H$ the dimension of each head, and $L$ the total number of layers. We denote by $\mathbf{X}_{\ell-1} \in \mathbb{R}^{t \times d}$ the residual stream entering layer $\ell$, with $\ell = 1, \ldots, L$.



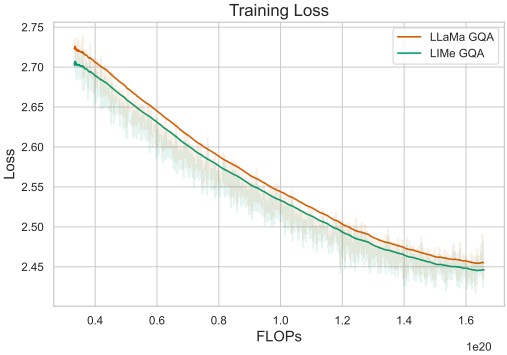

Figure 1: Training loss per FLOPs for LLaMa and LIMe. LIMe has a substantially lower loss with a similar amount of FLOPs. See Section 5.1 for more details.

**Causal Self-Attention.** Let
$$\mathbf{Q} = \mathbf{X}\,\mathbf{W}^{(Q)}, \quad \mathbf{K} = \mathbf{X}\,\mathbf{W}^{(K)}, \quad \mathbf{V} = \mathbf{X}\,\mathbf{W}^{(V)},$$
with $\mathbf{W}^{(Q)}, \mathbf{W}^{(K)}, \mathbf{W}^{(V)} \in \mathbb{R}^{d \times d}$. Splitting into $H$ heads of dimension $d_h = d/H$ yields $\{\mathbf{Q}_i, \mathbf{K}_i, \mathbf{V}_i\}_{i=1}^{H}$. For head $i$,
$$\text{head}_i = \text{softmax}\!\left(\frac{\mathbf{Q}_i\,\mathbf{K}_i^{\top}}{\sqrt{d_h}} + \mathbf{M}\right)\mathbf{V}_i \in \mathbb{R}^{t \times d_h},$$
where $\mathbf{M}$ masks future positions. The heads are concatenated across the last dimension and projected:
$$\text{MultiHeadAttn}(\mathbf{X}) = \text{Concat}(\text{head}_1, \dots, \text{head}_H)\,\mathbf{W}^{(O)}, \quad \mathbf{W}^{(O)} \in \mathbb{R}^{d \times d}.$$

**Residual connections.** Denoting a sub-layer function $\mathcal{F}(\cdot)$ and input $\mathbf{X}$, the pre-norm residual update is
$$\mathbf{X}' = \mathbf{X} + \mathcal{F}\big(\text{RMSNorm}(\mathbf{X})\big).$$

## 4 METHOD

We introduce *Layer-Integrated Memory* (LIMe), a lightweight mechanism to augment a decoder-only Transformer with inter-layer, learnable information flow. Unlike standard multi-head attention (MHA), which attends only to the current layer's residual stream, LIMe enables each head to retrieve and fuse Key–Value representations from all earlier layers. This enriches the model's representation capacity without increasing memory use, since we reuse the Key–Value buffers already allocated by vanilla Transformers.

At a high level, each LIMe attention layer performs three steps:

1. Compute and *buffer* per-head Key–Value projections from the current residual stream.
2. *Route* by forming a learned mixture of all buffered Key and Value heads' states up to the current layer.
3. Compute attention between the current layer's Queries and the routed Key–Value mixture.

Visualisation of the architecture can be found in Appendix K.

**1. Key–Value Buffering.** At layer $\ell$, we compute per-head Key and Value tensors in the usual way:
$$\mathbf{K}_\ell = \mathbf{X}_{\ell-1}\,W_\ell^{(K)}, \quad \mathbf{V}_\ell = \mathbf{X}_{\ell-1}\,W_\ell^{(V)}, \quad \mathbf{K}_\ell, \mathbf{V}_\ell \in \mathbb{R}^{t \times H \times d_h}. \tag{1}$$
We then store these in the pre-allocated buffers
$$\mathcal{B}^{(K)}, \mathcal{B}^{(V)} \in \mathbb{R}^{L \times H \times t \times d_h},$$
for Keys and Values respectively. No extra memory is required, since vanilla Transformers already maintain all per-layer Key–Value states for training and cache them during inference for generation efficiency. See Appendix I for details.

**2. Inter-Layer Routing.** To enable each head at layer $\ell$ to *mix* information from all previous layers, we introduce a trainable router tensor $R^{(\ell)} \in \mathbb{R}^{\ell \times H \times H}$, where $R^{(\ell)}_{\ell',h',h}$ is a weight from head $h'$ at layer $\ell'$ into head $h$ at layer $\ell$.

Using buffer we route keys and values for each head $h$:

$$\widetilde{\mathbf{K}}_{\ell,h} = \sum_{\ell'=1}^{\ell} \sum_{h'=1}^{H} R^{(\ell)}_{\ell',h',h} \, \mathcal{B}^{(K)}_{\ell',h'}, \quad \text{and} \quad \widetilde{\mathbf{V}}_{\ell,h} = \sum_{\ell'=1}^{\ell} \sum_{h'=1}^{H} R^{(\ell)}_{\ell',h',h} \, \mathcal{B}^{(V)}_{\ell',h'}. \tag{2}$$

**3. Attention with Layer-Integrated Memory.** We compute the usual per-head Queries,

$$\mathbf{Q}_{\ell,h} = \mathbf{X}_{\ell-1} \, W^{(Q)}_{\ell,h}, \quad \mathbf{Q}_{\ell,h} \in \mathbb{R}^{t \times d_{\mathrm{h}}},$$

and then perform scaled dot-product attention for each head between $\mathbf{Q}_{\ell,h}$ and the routed $\widetilde{\mathbf{K}}_{\ell,h}, \widetilde{\mathbf{V}}_{\ell,h}$.

**LIMe Advantages.** By routing through all prior layers, LIMe endows each head with a learnable, layer-wise memory. Unlike fixed skip connections or naive averaging, LIMe learns per-head, per-layer weightings, enabling selective retrieval and *forgetting* of past representations. Despite this added flexibility, the extra computation is only linear in sequence length. Crucially, LIMe is fully compatible with efficient MHA implementations such as FlashAttention (Dao, 2024), and it introduces negligible additional memory footprint by reusing existing Key–Value buffers (see Appendix I for details), and can be effectively used under pipeline parallelism (see Appendix J for details). In Appendix F, we include an ablation study on restricted router weights, demonstrating the importance of the trained router in LIMe.

## 5 EXPERIMENTS

### 5.1 LANGUAGE MODELING

We evaluate the effectiveness of **LIMe** against three baselines: **LLaMa** (Grattafiori et al., 2024), **DenseFormer** (Pagliardini et al., 2024), and **Hyper Connections** (Zhu et al., 2024). All models have approximately 1B parameters and share the same underlying transformer architecture (see Table 4). We trained each model from scratch on the *FineWeb Edu* (Penedo et al., 2024) subset with about 50B tokens. The full training setup can be found in Appendix A.

Figure 1 displays the iso-flops training loss curves, demonstrating that LIMe converges more rapidly and achieves lower perplexities than LLaMa, indicating improved parameter efficiency. Details on model efficiency and FLOPs calculations can be found in Appendix I. Table 1 presents results on the 3-shot LM Eval Harness benchmarks Wang et al. (2018; 2019); Srivastava et al. (2023), further highlighting the advantages conferred by LIMe on language modeling over baseline models. For more benchmarks see Appendix C. In the next section, we go deeper into the factors driving these gains.

| Model | MultiRC | WiC | QNLI | ARC-E | ARC-C | KV | Induction | Avg |
|---|---|---|---|---|---|---|---|---|
| LLaMA | 43.24 | 50.00 | 49.49 | 70.45 | 38.70 | 45.94 | 54.20 | 50.29 |
| DenseFormer | 45.92 | 49.69 | 50.08 | 70.60 | 36.48 | 50.30 | 51.30 | 50.62 |
| HC | 54.34 | 49.72 | 49.43 | 71.15 | 37.63 | 51.68 | 51.59 | 52.22 |
| **LIMe** | **56.15** | **50.44** | **51.43** | **71.15** | **39.30** | **55.64** | **55.36** | **54.21** |

Table 1: LM Evaluation Harness benchmarks results on 1B models with GQA in 3-shot setup. LIMe outperforms LLaMA, DenseFormer, and Hyper-Connections baselines. See details in Section 5.1 and additional benchmarks in Appendix C.

### 5.2 MATH WORD PROBLEMS (GSM8K)

To assess multi-step numerical reasoning in natural language, we evaluate on GSM8K (Cobbe et al., 2021). We *fully fine-tune* both LLaMA and LIMe (training details in Appendix A). LIMe clearly out-

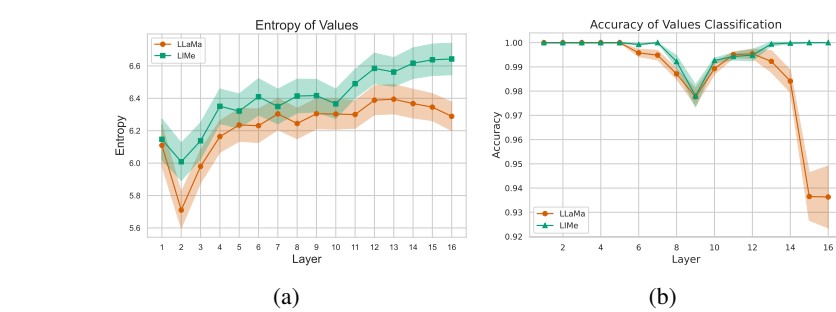

(a)  (b)

Figure 2: (a) Matrix entropy of values on the FineWeb Edu subset by layer. LIMe has more diverse values than LLaMa, which indicates that more information is stored in its hidden states. (b) Values' classification accuracy, with standard deviation over five cross-validation folds. Values in later layers obtained from LIMe can be linearly separated with nearly 1.0 accuracy, whereas the accuracy for values from LLaMa is much lower. See Section 5.3 for more details.

performs LLaMA, achieving an exact-match accuracy of **0.167** vs. **0.140** for LLaMA— a **+19.28%** relative improvement.

## 5.3 MEASURING REPRESENTATION COLLAPSE

Recent work has shown that large language models (LLMs) can suffer from *representation collapse* when representing long sequences, thereby forcing subtle token distinctions to become inseparable in deeper layers (Voita et al., 2019; Arefin et al., 2024). We investigate this phenomenon by comparing LLaMa (Grattafiori et al., 2024) and LIMe via two complementary approaches: (i) quantifying the diversity of hidden states and values with *matrix-based Rényi entropy* (Arefin et al., 2024) and (ii) measuring and visualizing the linear separability of layer-wise embeddings of closely related tokens (is, are, was, were) (Voita et al., 2019). These two methodologies directly measure representation collapse in language models.

Unlike Arefin et al. (2024), we evaluate both residual-stream hidden states and value representations. We expect weaker linear separability in hidden states (because the model need not pack all information there) and stronger separation in value vectors. For matrix entropy, we anticipate little change at the hidden-state level but a clear difference for value representations. At each layer $\ell$, we record *value states* (i.e., the output of the $W_\ell^{(V)}$ linear projection) and *hidden states* (i.e., the residual stream $\mathbf{X}_\ell$).

**Matrix-Based Rényi Entropy.**  Following Arefin et al. (2024), we measure the diversity of representations at layer $\ell$ by forming the Gram matrix $\mathbf{K} = Z^{(\ell)} Z^{(\ell)^\top} \in \mathbb{R}^{t \times t}$, where $Z^{(\ell)}$ contains the $d$-dimensional representations of $t$ tokens. Let $\{\lambda_i(\mathbf{K})\}_{i=1}^t$ be the eigenvalues of $\mathbf{K}$. We define the $\alpha$-order Rényi entropy as $S_\alpha\big(Z^{(\ell)}\big) = \frac{1}{1-\alpha} \log\left[ \sum_{i=1}^t \left(\frac{\lambda_i(\mathbf{K})}{\mathrm{tr}(\mathbf{K})}\right)^\alpha \right]$. Each eigenvalue is normalized by $\mathrm{tr}(\mathbf{K})$, ensuring the probabilities sum to 1. Higher $S_\alpha$ indicates greater variance (i.e., lower collapse).

Figure 2(a) shows that LIMe yields significantly higher matrix entropy of gathered MHA values compared with LLaMa and shows no significant difference when evaluating hidden states (see Figure 7(a)).

**Layer-Wise Token Separability.**  To more directly evaluate the level of representation collapse, we replicate the methodology of Voita et al. (2019), extracting 1668 occurrences each of is, are, was, were from the *FineWeb Edu* corpus. To quantify information collapse, we train a linear four-way classifier (for is, are, was, were) on layer-wise representations. Figure 2(b) shows mean classification accuracies (with five-fold cross-validation) for value representations layer by layer. We observe that LIMe consistently exhibits higher classification accuracy than LLaMa, confirming that LIMe's value representations avoid collapse. As hypothesized, hidden states became less separable

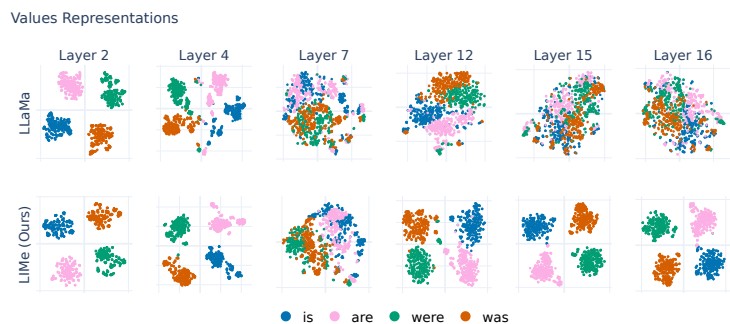

Figure 3: t-SNE of similar tokens' values among layers shows higher separability for LIMe's representations. See Section 5.3 for more details.

for LIMe, indicating that there was no need to store all necessary information in a single hidden state (see Figure 7(b)).

Additionally, we project representations into a two-dimensional space via t-SNE and visualize how well value states and hidden states can be clustered (Figure 3). In contrast to LIMe, deeper-layer representations in LLaMa for such similar tokens often collapse into overlapping regions, reflecting the inclination of the vanilla transformer to heavily compress relevant information into a single representation and therefore blur small yet important differences.

**Linear Probing.** We evaluate whether layer-wise representations encode basic grammaticality using BLiMP (Warstadt et al., 2020). For each BLiMP task, we freeze the LM and train a *binary* logistic-regression probe that predicts whether a *single sentence* is grammatical ("Good") or ungrammatical ("Bad"). Concretely, at each layer $\ell$ we extract (i) *attention values* (the value projections) and (ii) *hidden states* (the residual stream), mean-pool them over tokens to obtain a fixed vector per sentence, and fit a logistic regression on these vectors. We perform 5-fold cross-validation, splitting by minimal pair so that both members of a pair fall in the same fold, and report accuracy in Table 2. At test time the probe receives one sentence and outputs a grammaticality label; accuracy is the fraction of correct Good/Bad judgments.

| Layer | Values (acc.) | | Hiddens (acc.) | |
|---|---|---|---|---|
| | **LLaMA** | **LIMe** | **LLaMA** | **LIMe** |
| 10 | $0.892 \pm 0.018$ | $\mathbf{0.914} \pm 0.015$ | $0.914 \pm 0.015$ | $\mathbf{0.933} \pm 0.013$ |
| 14 | $0.881 \pm 0.015$ | $\mathbf{0.921} \pm 0.013$ | $0.895 \pm 0.015$ | $\mathbf{0.918} \pm 0.014$ |
| 16 | $0.864 \pm 0.016$ | $\mathbf{0.918} \pm 0.010$ | $0.880 \pm 0.016$ | $\mathbf{0.897} \pm 0.014$ |

Table 2: BLiMP probing accuracy (5-fold CV) at selected layers (for complete results see Appendix D). LIMe consistently outperforms LLaMA, with gains up to 5 p.p. on value features and 3 p.p. on hidden states, indicating more linearly separable (and thus more expressive) representations.

**Discussion.** Together, these results corroborate our theoretical motivation: by allowing each head to attend directly to earlier-layer representations, LIMe expands the overall representational capacity. This multi-layer routing reduces collapse in the *values* while freeing deeper *hidden* states from the burden of storing all lexical nuances—leading to higher overall entropy on values (Figure 2(a)) and improved model performance (Table 1). In the next section, we evaluate LIMe on synthetic benchmarks where the model's ability to store complex information in limited state capacity is crucial.

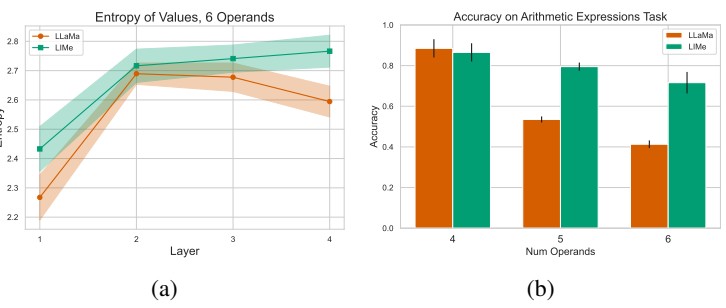

(a)                                             (b)

Figure 4: (a) LIMe exhibits consistently higher entropy of value vectors across layers, particularly in the final layer, indicating reduced representation collapse compared to LLaMa. (b) On the Arithmetic Expressions task, LIMe significantly outperforms the LLaMa baseline, maintaining high accuracy even as the number of operands increases, while LLaMa's performance deteriorates. For details, see Section 5.4.2.

## 5.4 EVALUATING REPRESENTATION COLLAPSE ON SYNTHETIC TASKS

### 5.4.1 PLANNING AND SEARCH CAPABILITIES

We fine-tune models on ProsQA (Proof with Search Question-Answering) (Hao et al., 2024). Each ProsQA instance presents a set of fictional concepts described via natural-language conditions arranged in a DAG, requiring models to determine the veracity of a target statement by exploring multiple reasoning paths over the graph (examples in Appendix B). Unlike linear chain-of-thought methods (Wei et al., 2022), ProsQA demands maintaining and evaluating parallel hypothesis streams akin to breadth-first search in latent reasoning (Hao et al., 2024). In our experiments we evaluate both fine-tuned models on ProsQA task via open-ended reasoning generation. LLaMA achieves **69.4%** accuracy, meanwhile LIMe achieves **77.8%** accuracy, outperforming LLaMA by **8.4%**. Since correct prediction requires searching over paths in the graph of input statements, baseline transformers suffer representation collapse from storing multiple reasoning chains in their hidden states, particularly for longer inference sequences. LIMe mitigates this by distributing the reasoning process across layers — early layers may store primitive inferences while deeper layers compose them, maintaining better separation between similar reasoning paths.

### 5.4.2 ARITHMETIC EXPRESSION BENCHMARK

Standard one-shot QA benchmarks mainly test *final-token prediction*, which can often be solved via shallow pattern matching or retrieval, masking the role of intermediate representation quality in reasoning. To isolate the impact of multi-step computation, we adopt the Arithmetic Expression Task (AET) (Arefin et al., 2024; Feng et al., 2023), a synthetic benchmark presenting expressions over integer operands with operators $+, -, \times, \div$, along with solution steps and requiring the exact integer result. See examples in Appendix B.

Following Arefin et al. (2024), we generate 3 difficulty tiers comprising expressions with 4, 5, and 6 operands, accompanied by step-by-step solutions (details in Appendix A). While performing similarly to LLaMa on 4 operands, LIMe achieves significantly higher accuracy after increasing number of operands to 5 and 6 (Figure 4(b)). LIMe (**71.6%**) outperforms LLaMa (**41.3%**) by over **30%** in accuracy on 6 operands. These results go along with lower representation collapse which is illustrated by higher entropy of value representations shown in Figure 4(a). Also, LIMe exhibits better separability of close numbers which leads to lower error rate in intermediate calculations, see Figure 8 in Appendix.

Arithmetic Expressions Task requires intermediate calculations to be performed correctly in order to get the correct final answer. The problem of representation collapse results in representations of close numbers being similar which leads to incorrect intermediate results, and thus the wrong final answer. Since LIMe has access to previous representations at each layer, it preserves finer numerical distinctions in comparison with standard transformer architectures like LLaMa. Moreover, LIMe has ability to store information in earlier representations, i.e. performing computations at some

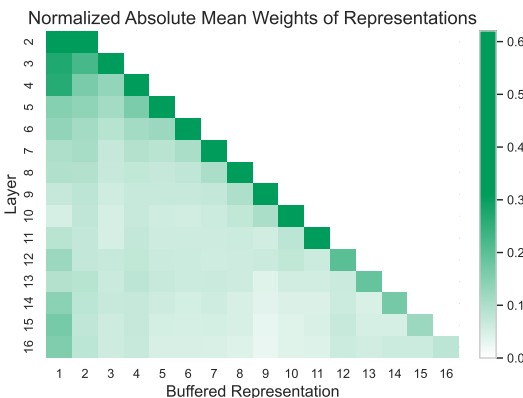

Figure 5: Mean retrieval weight for each buffered representation across subsequent layers. Larger diagonal values confirm reliance on the current residual stream, while the pronounced off-diagonal weights for the earliest buffers and the repeated reuse of intermediate ones show that the model systematically retrieves earlier features, providing auxiliary memory and helping to mitigate representation collapse. See Section 5.5 for more details.

early or intermediate layer, but using it further only in later layers, which also boosts its reasoning capabilities and leads to better results on tasks that require intermediate steps.

## 5.5 ANALYZING LEARNED ROUTINGS IN LIMe

To understand *how* LIMe routes information across layers and thereby mitigates representation collapse, we inspect the learned router weights. Since the router weights can be both positive and negative—and because random initialization of the key, value, and output projections renders their sign semantically ambiguous—we analyze the absolute magnitudes of these weights to quantify each buffered representation's relative contribution in a sign-agnostic manner.

For each layer $\ell \geq 2$, we take the absolute magnitude of its router weights, average over heads for each buffered representation $j \leq \ell$, and then normalize these averages per layer. The resulting heatmap in Figure 5 shows the normalized mean weight: cell $(\ell, j)$ measures the average contribution of the keys and values generated at layer $j$ to the attention computation in layer $\ell$. In a standard Transformer without routing, each layer would attend solely to its own keys and values, yielding a heatmap with ones on the diagonal and zeros elsewhere; LIMe departs markedly from this behavior.

Several clear patterns emerge:

- **Strong reliance on embeddings in early layers:** Layers 2-4 allocate much of their attention to the buffered representations from the embedding layer. This corroborates the view that the initial attention layers focus on capturing local and morphological relationships among tokens, and that LIMe grants additional flexibility in reusing these low-level features.

- **Auxiliary memory via neighboring layers:** Early and middle layers place a share of attention on the buffered KV states of its immediate predecessor. This indicates that they can treat them as an auxiliary memory bank, effectively extending the subspace of features it can manipulate by leveraging projections made by other heads.

- **Long-distance retrieval from early buffers:** Higher layers also attend nontrivially to the first two buffered representations. The effect is especially pronounced in the final layers, suggesting that late-stage prediction benefits from revisiting the original token embeddings and shallow features.

By allowing flexible retrieval of features from arbitrarily distant layers, LIMe relieves each residual stream from having to carry the entire contextual signal forward. Instead, information can be distributed across a set of persistent buffers, preserving a richer and more diverse feature set through-

out the network's depth and thereby mitigating representation collapse. For the full, detailed set of normalized router weights, see Appendix Figure 9.

## 5.6 DEEP NETWORKS PERFORMANCE

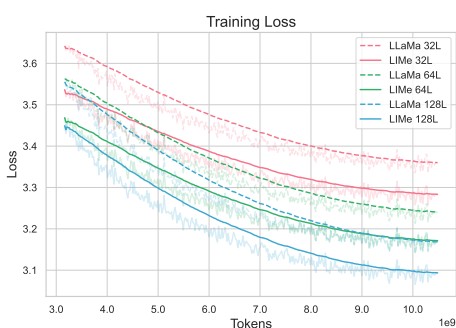

Figure 6: Training losses for deep architectures. The LIMe models consistently outperform their LLaMA counterparts across all depths, with LIMe with 64 layers outperforming LLaMa with 128 layers. See Section 5.6 for details.

Transformers scaled to increasing depths often suffer from representation collapse, which motivates our evaluation of LIMe in 32-, 64-, and 128-layer configurations. We compare LIMe against the baseline LLaMA, each using 8 attention heads per layer, and observe that LIMe outperforms LLaMA at every tested depth (Fig. 6). Furthermore, LIMe exhibits superior scaling behavior: as depth increases, its loss decreases more rapidly than LLaMA's, implying that direct routing of earlier-layer features enhances the model's effective representational capacity, whereas LLaMA's single-stream residual architecture struggles to preserve fine-grained features across layers. Notably, a 64-layer LIMe model outperforms a 128-layer LLaMA model, despite the latter requiring roughly twice the FLOPs and parameters. In the 128-layer regime, the naive LIMe router that mixes all previous layers yields a substantial perplexity reduction over LLaMA but introduces a noticeable per-step latency increase. However, simpler structured routers (such as dilated routing and variants that restrict each layer to attend only to the set of $j$ earliest layers) incur only negligible latency overhead and essentially no extra memory while still achieving significantly better perplexity than the 128-layer LLaMA baseline (see Appendix F for details). This suggests that the optimal scaling strategy for transformers may deviate from conventional practice, potentially favoring much deeper models with smaller hidden dimensions. We leave further investigation of these scaling dynamics to future work.

## 6 CONCLUSION AND FUTURE WORK

In this paper, we proposed *Layer-Integrated Memory* (LIMe), a lightweight extension to multi-head self-attention that enables each attention head to retrieve and integrate representations from all preceding layers. Through extensive experiments on language modeling, synthetic reasoning benchmarks, and deep transformer configurations, we demonstrated that LIMe (i) accelerates convergence in FLOPs by up to 15.3% and reduces perplexity by up to 1.15% compared to standard Transformer decoders, yields improvements of up to +8% on the challenging ProsQA task and +30% on Arithmetic Reasoning Task; (ii) mitigates representation collapse by preserving higher entropy in value vectors and maintaining token separability in deeper layers; and (iii) enables shallower models to match or exceed the performance of double-sized deeper baselines. Our analysis of the learned routing weights further revealed that LIMe systematically leverages both local and long-distance feature reuse, effectively distributing contextual information across layers without increasing the hidden-state size.

**Limitations.** While our method consistently yields better results on both benchmarks and language modeling tasks, it could lead to additional communication between GPUs in pipeline parallel setup. Also, vanilla implementation of the method has $\mathcal{O}(L^2)$ asymptotic, and some heuristics proposed in Appendix F might be useful for scaling.

Looking forward, two research directions emerge as particularly promising. First, a comprehensive exploration of the width–depth trade-off in LIMe architectures could unveil optimal scaling regimes tailored to diverse tasks and computational budgets. Second, a rigorous theoretical analysis of the routing mechanism may inform principled designs for multi-layer memory, thereby enabling models to perform advanced latent-space reasoning grounded in Layer-Integrated Memory.

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

## A EXPERIMENTAL SETUP DETAILS

**Language Modeling.** We observe that omitting weight decay on the LIMe router weights enjoys better performance and setting the router's learning rate to $1 \times 10^{-2}$ boosts model performance by speeding up router convergence and circuit formation. To preserve the standard Transformer's information flow at the start of the training, we initialize the slice $R^{(\ell)}_{\ell,h',h} = \delta_{h',h}$ (identity across heads). Other coefficients are initialized randomly via Kaiming uniform to stabilize mixtures at the start of the training. Random initialization of all weights resulted in worse overall model performance. For DenseFormer and HyperConnections baselines we use the strongest configurations recommended by the original papers: DenseFormer with dilation = 1 and period = 1, and the Dynamic HyperConnections variant with expansion rate 4. Hyperparameter values are summarized in Table 3, and the detailed model architecture is given in Table 4. Additional training loss visualizations are available in Figure 11 for full attention and in Figure 10 for Grouped Query Attention.

We used NVIDIA H100 GPUs and spent about 2400 GPU-days on all experiments including preliminary research.

**GSM8K Fine-tuning.** We fine-tune pretrained 1.2B-parameter LLaMa and LIMe models on the GSM8K training split for 20 epochs and report exact-match accuracy on the test set. Learning rates

are tuned per model for best performance—$1 \times 10^{-4}$ for LLaMa and $5 \times 10^{-5}$ for LIMe—with an effective batch size of 32 in both cases.

**ProsQA Fine-Tuning.** We fine-tune pretrained LLaMa 150M and LIMe 150M on approximately 18,000 sequences for 10 epochs. We use learning rate of $1 \times 10^{-4}$ with linear decay and warmup during the first epoch, effective batch size is 128. Trained models are then evaluated on the test subset via open generation of reasoning steps and answers.

**Arithmetic Expression Task.** We train models and evaluate them on open-ended generation of solutions given initial expression, from which we extract the answers and calculate accuracy on the test subset. We train 4-layer models (with 4 attention heads and model dim is 32) on datasets with 50,000 samples per each number of operands for 200 epochs. Learning rate is $1 \times 10^{-3}$ with linear decay.

| Hyperparameter | Value |
|---|---|
| Optimizer | AdamW |
| Learning Rate | 0.001 |
| LIMe Router Learning Rate | 0.01 |
| Weight Decay | 0.1 |
| $\beta_1$ | 0.9 |
| $\beta_2$ | 0.95 |
| $\epsilon$ | $1 \times 10^{-8}$ |
| Scheduler | cosine |
| Warmup Steps | 200 |
| Min LR | $1 \times 10^{-6}$ |
| Mixed Precision | bf16 |
| Gradient Clipping | 1.0 |
| Sequence Length | 2048 |
| Batch Size | 1024 |
| Training Steps | 20,000 |

Table 3: Key training hyperparameters used in experiments.

| Parameter | Value |
|---|---|
| Vocab Size | 50,257 |
| Hidden Size | 2048 |
| Intermediate Size | 8192 |
| Number of Hidden Layers | 16 |
| Number of Attention Heads | 32 |
| Number of Key-Value Heads | 8 (GQA) and 32 (otherwise) |
| Tie Word Embeddings | True |

Table 4: Base model architecture at 1B scale.

## B  SYNTHETIC BENCHMARKS

---

**ProsQA**

Question: "Every shumpus is a rempus. Every shumpus is a yimpus. Every terpus is a fompus. Every terpus is a gerpus. Every gerpus is a brimpus. Alex is a rempus. Every rorpus is a scrompus. Every rorpus is a yimpus. Every terpus is a brimpus. Every brimpus is a lempus. Tom is a terpus. Every shumpus is a timpus. Every yimpus is a boompus. Davis is a shumpus. Every gerpus is a lorpus. Davis is a fompus. Every shumpus is a boompus. Every shumpus is a rorpus. Every terpus is a lorpus. Every boompus is a timpus. Every fompus is a yerpus. Tom is a dumpus. Every rempus is a rorpus. Is Tom a lempus or scrompus?"

Steps: "Tom is a terpus. Every terpus is a brimpus. Every brimpus is a lempus."

Answer: "Tom is a lempus."

---

**Arithmetic Expression Task**

**Input:**

$$(7 + 5) \div (6 + 4 \times 3 - 2 \times 7) =$$

**Output:**

$$12 \div (6 + 4 \times 3 - 2 \times 7) = 12 \div (6 + 12 - 2 \times 7)$$
$$= 12 \div (18 - 2 \times 7)$$
$$= 12 \div (18 - 14)$$
$$= 12 \div 4$$
$$= 3$$

---

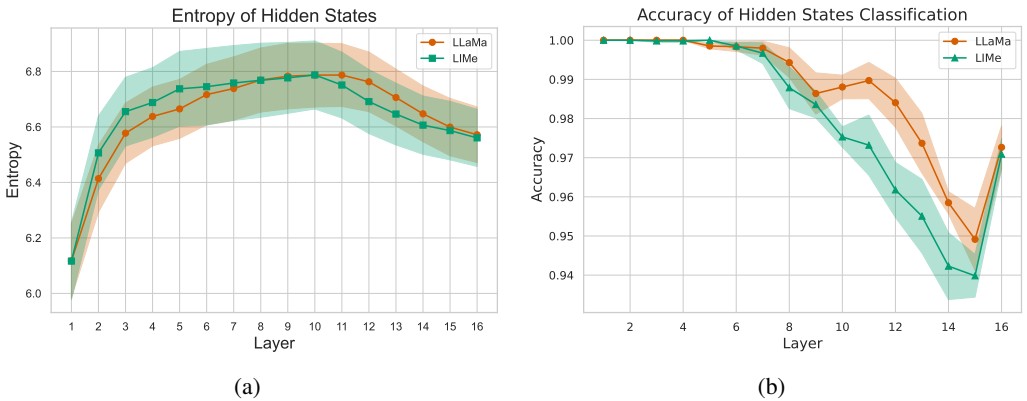

(a)                                          (b)

Figure 7: (a) Matrix entropy of the hidden states across layers on the FineWeb Edu subset. We do not observe a significant difference between LIMe and LLaMa in this experiment. (b) Classification accuracy of the hidden states, with standard deviation, measured over five cross-validation folds. Because the hidden states in LIMe do not need to store all the information in the residual stream, they become less separable. See Section 5.3 for more details.

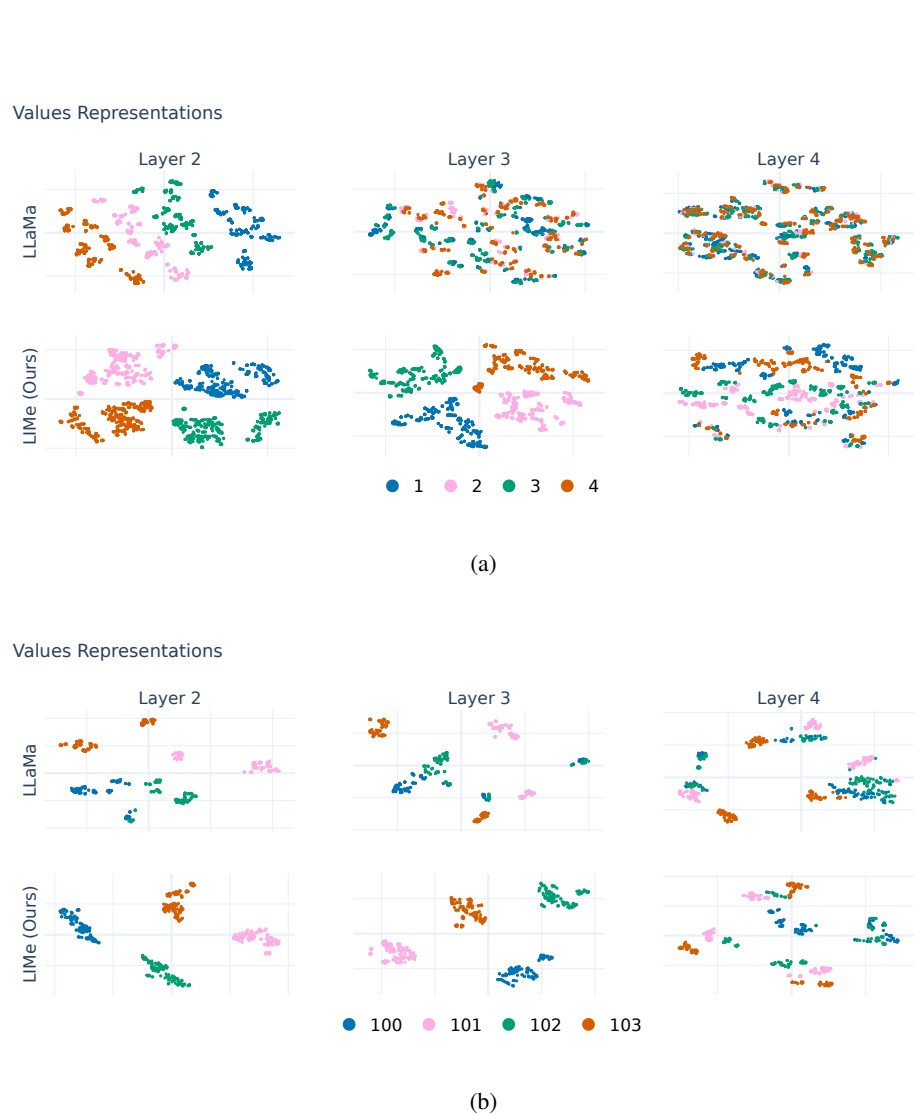

Figure 8: t-SNE of close numbers' values representations of models trained on Arithmetic Expressions Task. (a) For $1, 2, 3, 4$. (b) For $100, 101, 102, 103$. See Section 5.4.2.

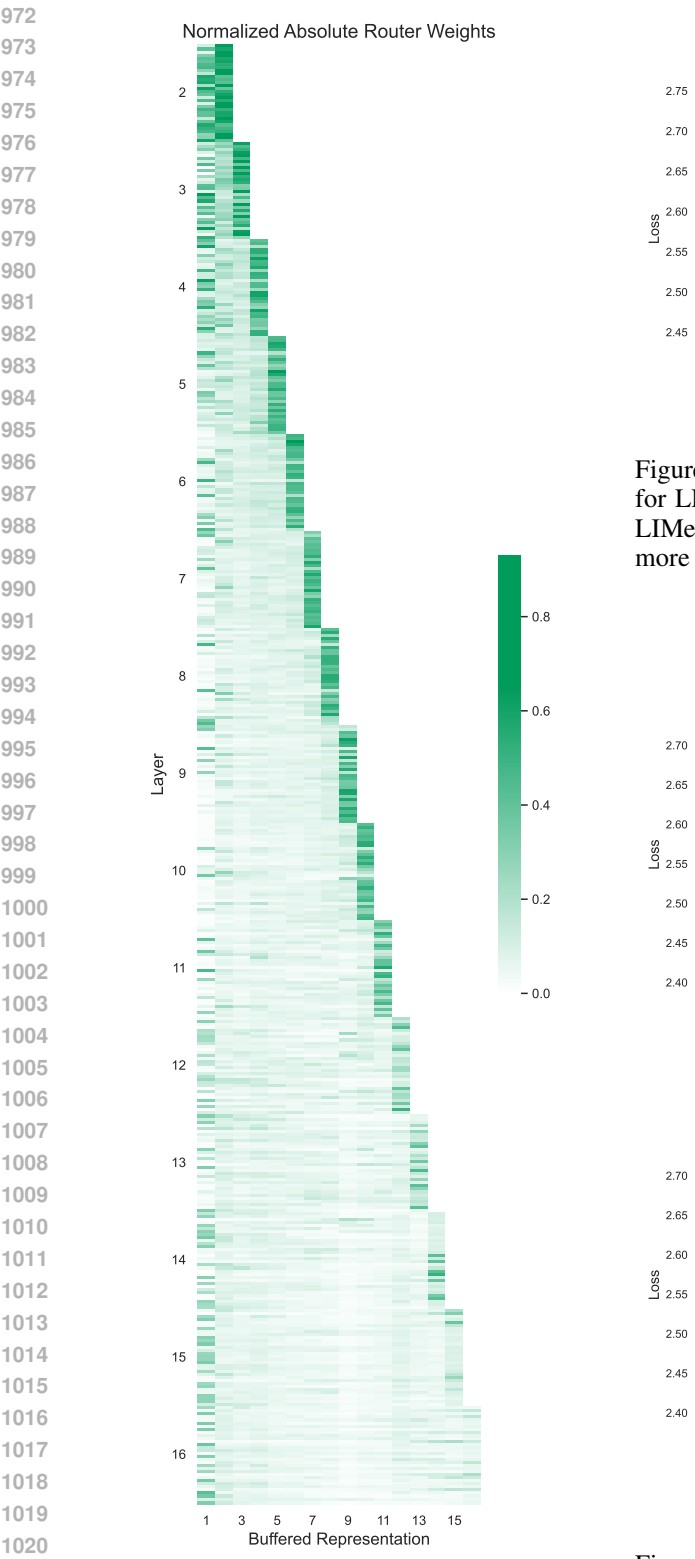

Figure 9: Magnitudes of router weights averaged among buffered heads and normalized among buffered layers. Each cell represents ratio of attention for each buffered representation in the specific head.

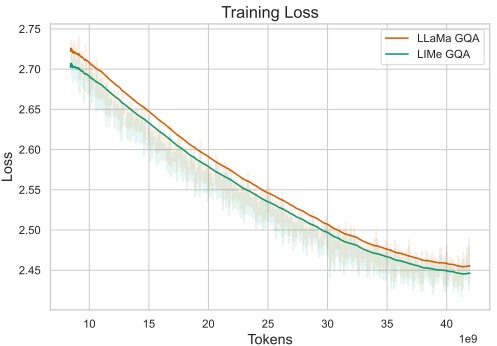

Figure 10: Training loss per tokens trained on for LLaMa and LIMe with GQA. It shows that LIMe is more data efficient. See Section 5.1 for more details.

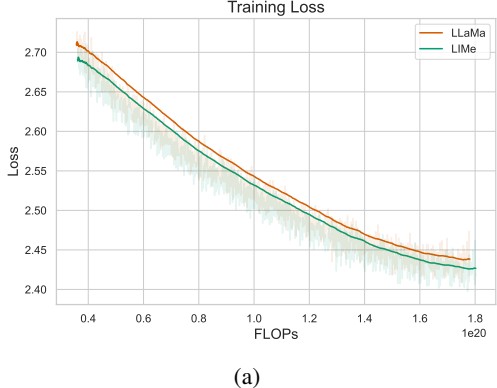

(a)

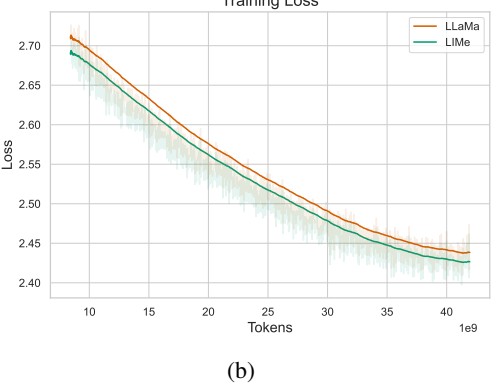

(b)

Figure 11: Training loss for LLaMa and LIMe without GQA. (a) shows that LIMe has a substantially lower loss with a similar amount of FLOPs. (b) shows that LIMe is more data efficient. See Section 5.1 for more details.

## C  ADDITIONAL BENCHMARKS

| Model | COPA (50) | MultiRC (50) | WiC (50) | QNLI (50) | WNLI (50) | Avg (50) |
|---|---|---|---|---|---|---|
| LLaMA | **75.80**±1.92 | 43.24±0.32 | 50.00±0.89 | 49.49±0.30 | 51.27±2.66 | 53.96 |
| DenseFormer | 74.00±1.96 | 45.92±0.32 | 49.69±0.89 | 50.08±0.30 | 52.11±2.66 | 54.36 |
| HC | 74.00±1.96 | 54.34±0.32 | 49.72±0.89 | 49.43±0.30 | **56.34**±2.64 | 56.77 |
| **LIMe** | 75.20±1.93 | **56.15**±0.32 | **50.44**±0.89 | **51.43**±0.30 | 56.06±2.64 | **57.86** |

Table 5: GLUE and SuperGLUE benchmarks accuracies (%) on 1B GQA models (3-shot), with average over the five tasks. Random baselines in parentheses.

| Model | ARC-E (25) | ARC-C (25) | HellaSwag (25) | OBQA (25) | Avg (25) |
|---|---|---|---|---|---|
| LLaMA | 70.45±0.42 | 38.70±0.64 | 52.55±0.22 | 37.68±0.97 | 49.85 |
| DenseFormer | 70.60±0.42 | 36.48±0.63 | 41.46±0.22 | 26.84±0.89 | 43.85 |
| HC | 71.15±0.42 | 37.63±0.63 | **54.04**±0.22 | **40.08**±0.98 | 50.73 |
| **LIMe** | **71.15**±0.42 | **39.30**±0.64 | 52.85±0.22 | 39.68±0.98 | **50.75** |

Table 6: QA benchmarks accuracies (%) on 1B GQA models (3-shot), with average over the four tasks. Random baselines in parentheses.

| Model | KV (50) | Induction (50) | IR (0.04) | CO (0.06) | Avg (25.03) |
|---|---|---|---|---|---|
| LLaMA | 45.94±2.22 | 54.20±2.69 | 12.94±1.63 | 16.97±0.38 | 32.51 |
| DenseFormer | 50.30±2.23 | 51.30±2.69 | **15.76**±1.77 | **18.59**±0.39 | 33.99 |
| HC | 51.68±2.23 | 51.59±2.69 | 15.29±1.75 | 18.48±0.39 | 34.26 |
| **LIMe** | **55.64**±2.21 | **55.36**±2.68 | 14.82±1.73 | 17.39±0.38 | **35.80** |

Table 7: Accuracies (%) of 3-shot 1B GQA models on BIG-Bench tasks: Key–Value Maps (KV), Mathematical Induction, Implicit Relations (IR), and Reasoning About Colored Objects (CO). Random baselines in parentheses.

## D  LINEAR PROBING RESULTS

We evaluate linguistic sensitivity using ten BLiMP minimal-pair tasks (Warstadt et al., 2020). For each task, we use a representative pair (Good/Bad) to illustrate the contrast; full datasets are from the public BLiMP repository. Below, the numbered list (1–10) gives task names, and the Table 8 maps each task to a representative example.

1. Determiner–Noun Agreement with Adjective (Irregular), set 1

2. Complex NP Island

3. Subject–Verb Agreement with Regular Plurals, set 2

4. Determiner–Noun Agreement with Adjective, set 2

5. Determiner–Noun Agreement, set 1

6. Determiner–Noun Agreement, set 2

7. Subject–Verb Agreement with Irregular Plurals, set 1

8. Subject–Verb Agreement with Irregular Plurals, set 2

9. Agreement with Distractor (Relational Noun)

10. Determiner–Noun Agreement with Adjective, set 1

| # | Good | Bad |
|---|------|-----|
| 1 | Some waiters broke this lost foot. | Some waiters broke this lost feet. |
| 2 | Who aren't most hospitals that hadn't talked about most waitresses alarming? | Who aren't most waitresses alarming most hospitals that hadn't talked about? |
| 3 | The students perform. | The student perform. |
| 4 | Cynthia scans these hard books. | Cynthia scans this hard books. |
| 5 | Raymond is selling this sketch. | Raymond is selling this sketches. |
| 6 | Some dog stunned this committee. | Some dog stunned these committee. |
| 7 | Those radii have scared that teenager. | Those radii has scared that teenager. |
| 8 | The women meet. | The woman meet. |
| 9 | A niece of most senators hasn't descended most slopes. | A niece of most senators haven't descended most slopes. |
| 10 | Rebecca was criticizing those good documentaries. | Rebecca was criticizing those good documentary. |

Table 8: Representative BLiMP minimal pairs (one per task). Row numbers 1–10 correspond to the task names listed above.

| Layer | Values (acc.) | | Hiddens (acc.) | |
|-------|---------------|---------------|----------------|----------------|
| | **LLaMA** | **LIMe** | **LLaMA** | **LIMe** |
| 10 | $0.892 \pm 0.018$ | $\mathbf{0.914} \pm 0.015$ | $0.914 \pm 0.015$ | $\mathbf{0.933} \pm 0.013$ |
| 11 | $0.892 \pm 0.016$ | $\mathbf{0.912} \pm 0.016$ | $0.912 \pm 0.013$ | $\mathbf{0.929} \pm 0.012$ |
| 12 | $0.889 \pm 0.013$ | $\mathbf{0.925} \pm 0.012$ | $0.908 \pm 0.012$ | $\mathbf{0.930} \pm 0.015$ |
| 13 | $0.883 \pm 0.016$ | $\mathbf{0.921} \pm 0.016$ | $0.903 \pm 0.013$ | $\mathbf{0.926} \pm 0.015$ |
| 14 | $0.881 \pm 0.015$ | $\mathbf{0.921} \pm 0.013$ | $0.895 \pm 0.015$ | $\mathbf{0.918} \pm 0.014$ |
| 15 | $0.871 \pm 0.016$ | $\mathbf{0.924} \pm 0.011$ | $0.886 \pm 0.014$ | $\mathbf{0.910} \pm 0.012$ |
| 16 | $0.864 \pm 0.016$ | $\mathbf{0.918} \pm 0.010$ | $0.880 \pm 0.016$ | $\mathbf{0.897} \pm 0.014$ |

Table 9: BLiMP (Warstadt et al., 2020) probing accuracy (5-fold CV) across layers 10–16. LIMe improves both value and hidden representations.

# E  INPUT-DEPENDENT ROUTING

We additionally implemented a Dynamic LIMe variant, in which routing weights are generated by projecting the current hidden state (queries) against per-layer, per-head learnable keys. This yields a fully dynamic routing matrix of shape $H \times (L \cdot H)$. While more expressive, this variant introduced substantially higher parameter count, FLOPs, and memory consumption.

Moreover, in early experiments, it achieved marginally worse perplexity than the static LIMe variant. Given our core design objective of maximizing efficiency with minimal overhead, we have chosen to emphasize the static routing mechanism in the final version.

# F  ROUTER ABLATION

We conduct an ablation study to assess the importance of learning full per-layer, per-head router weights in LIMe. Specifically, we compare the standard LIMe routing against several constrained variants on the 150M-parameter model, evaluating their impact on perplexity:

- **Fixed Average (`average`)**: Aggregates all buffered Key–Value representations via a uniform average, without any learned head-specific weighting.

- **Recent–$j$ (`last-`$j$)**: Restricts each layer $\ell$ to attend only to the most recent $\min(\ell, j)$ buffered representations; router weights for these representations are learned.

- **Initial–$j$ (`first-`$j$)**: Restricts each layer $\ell$ to attend only to the first $\min(\ell, j)$ buffers plus the immediately preceding layer; router weights for these are learned.

In addition to constraining which *layers* can be routed (`last-`$j$ and `first-`$j$), we also ablate the structure of the router weights themselves. In particular, we ask whether LIMe benefits primarily from mixing information *across heads*, or whether it is sufficient to restrict routing to the same head index across layers, and whether making the router more expressive at the per-dimension level improves performance.

| Model | Perplexity | Change to LIMe |
|---|---|---|
| LLaMA | 16.4611 | +3.36% |
| LIMe `average` | 16.4611 | +3.36% |
| LIMe `last-2` | 16.2810 | +2.22% |
| LIMe `last-4` | 16.1675 | +1.51% |
| LIMe `last-6` | 16.1351 | +1.31% |
| LIMe `first-2` | 15.9746 | +0.30% |
| LIMe `first-4` | 15.9586 | +0.20% |
| LIMe `first-6` | 15.9906 | +0.40% |
| **LIMe** | **15.9267** | — |

Table 10: Impact of constrained routing schemes on validation perplexity for the 150M-parameter model. Table reports perplexity for each scheme and the relative change with respect to the full LIMe model. The `average` variant fails to improve over the LLaMA baseline, indicating that uniform pooling of past representations is insufficient. Constraining attention to fixed windows of layers (`last-j` and `first-j`) yields modest gains but still underperforms the unrestricted router. By contrast, the full LIMe routing achieves the lowest perplexity (15.9267), corresponding to a 3.36% reduction relative to LLaMA, thereby confirming the necessity of learning full, per-head, per-layer router weights for optimal performance.

We therefore compare the default LIMe router against two additional variants on the same 150M setup:

- **No head mixing** (`no-head-mix`): each head in layer $\ell$ only mixes Key–Value states from the *same* head index across previous layers (router shape $[H, L]$ instead of $[H, L \cdot H]$). This removes all cross-head interactions in the router.
- **Per-dimension mixing** (`per-dim`): each previous head is weighted by a $d_{\text{head}}$-dimensional vector instead of a scalar (router shape $[H, L \cdot H \cdot d_{\text{head}}]$), making the router strictly more expressive and increasing the number of routing parameters by a factor of $d_{\text{head}}$.

| Setup | Loss | Perplexity |
|---|---|---|
| LLaMA | 2.80043 | 16.45 |
| LIMe (default) | 2.76889 | 15.94 (–3.1%) |
| LIMe `no-head-mix` | 2.83235 | 16.99 (+3.3%) |
| LIMe `per-dim` | 2.77911 | 16.10 (–2.1%) |

Table 11: Router-structure ablation at 150M scale. The `no-head-mix` variant restricts routing to the same head index across layers and removes cross-head interactions; it not only eliminates LIMe's gains but performs worse than the LLaMA baseline. The `per-dim` variant uses per-dimension router weights and is strictly more expressive (and more expensive) than the default scalar per-head router, yet remains worse than default LIMe.

Two conclusions follow. First, *mixing across heads is crucial*: the `no-head-mix` variant, which only aggregates the same head across layers, degrades perplexity to 16.99 (+3.3% vs. LLaMA), indicating that LIMe's benefit comes from cross-head interactions across layers rather than merely accessing deeper same-head features. Second, *per-dimension routing does not help in this regime*: although `per-dim` improves over LLaMA (16.10 vs. 16.45), it is still worse than the much simpler scalar per-head router (15.94), while introducing on the order of $d_{\text{head}}$ more routing parameters and higher cost. This suggests that a lightweight per-head scalar router is sufficient and more effective under our training budget, reinforcing the design choice used in the main experiments.

# G ROUTING VARIANTS IN DEEP MODELS

The ablations in Appendix F study constrained routing schemes at 150M scale. Here we complement them with a deep 128-layer setup (see Section 5.6), where the naive LIMe router that mixes all previous layers has a more pronounced computational cost. We compare full LIMe to structured variants that sparsify the set of routed layers but keep the same overall architecture.

In addition to the 128-layer LLaMA baseline and full LIMe, we consider:

- **Dilated-$d$ (`dil-d`)**: each layer routes only to a sparsified set of previous layers with fixed dilation factor $d$ (e.g., every 8th or 16th layer), so that each layer sees roughly $L/d$ routed sources instead of all $L$.

- **First-$j$ (`first-j`, deep)**: each layer routes only to the first $j$ layers plus itself, reusing early, stable representations while ignoring later intermediate layers when forming the routed Key–Value mixture. In the deep setting we use $j \in \{7, 15\}$ for $L = 128$.

Table 12 reports per-iteration time, peak memory, and perplexity for the 128-layer configuration:

| Model | Time / iter (ms) | Peak Mem (MB) | Perplexity |
|---|---|---|---|
| LLaMA | 70.21 | 2054.26 | 23.73 |
| LIMe full | 80.85 (+15.2%) | 2062.38 (+0.4%) | 20.72 (−12.7%) |
| LIMe `dil-8` | 71.59 (+2.0%) | 2055.38 (+0.05%) | 21.61 (−8.9%) |
| LIMe `dil-16` | 71.57 (+1.9%) | 2054.88 (+0.03%) | 21.84 (−8.0%) |
| LIMe `first-7` | 71.79 (+2.3%) | 2054.85 (+0.03%) | 20.55 (−13.4%) |
| LIMe `first-15` | 72.69 (+3.5%) | 2055.76 (+0.07%) | 20.50 (−13.6%) |

Table 12: Routing variants for 128-layer models. Percentages are relative to the 128-layer LLaMA baseline. Full LIMe yields the largest perplexity improvement but also a noticeable increase in per-step time. Simpler structured routers (dilated and `first-j`) retain most or all of the perplexity gains while keeping latency overhead in the low single digits and memory essentially unchanged.

Several trends emerge. First, full LIMe significantly improves perplexity in the deep regime (from 23.73 to 20.72) but increases step time by about 15%. Second, the `first-7` and `first-15` variants achieve slightly *better* perplexity than full LIMe (down to 20.50) while increasing latency by only 2–3.5% and leaving peak memory virtually unchanged. Finally, the dilated variants `dil-8` and `dil-16` offer an intermediate trade-off: they reduce latency overhead to about 2% while still providing 8–9% perplexity reductions over LLaMA.

These observations align with the router-weight heatmaps in Fig. 5, where later layers place most of their mass on early buffers. In very deep models, forcing each layer to consider all $L$ previous layers can make the router partially adapt to noisy mid-layer states. Restricting routing to early layers (`first-j`) or to a sparse subset of layers (`dil-d`) effectively keeps the informative early Key–Value buffers while discarding less useful mid-layer signals, which explains why these structured variants match or slightly outperform full LIMe in perplexity while having negligible overhead.

## H  LIME PSEUDOCODE

```python
class KVBuffer:
    def __init__(self):
        self.mat = None  # [(layers_so_far * kv_h), 2 * b * t * hd]

    def add_(self, key_states, value_states):
        # key_states/value_states: (b, kv_h, t, hd)
        b, kv_h, t, hd = key_states.shape
        kv = torch.cat([key_states, value_states], dim=-1) # (b, kv_h, t, 2*hd)
        kv = kv.permute(1, 0, 2, 3).reshape(kv_h, b * t * 2 * hd) # (kv_h, b*t*2*hd)
        self.mat = kv if self.mat is None else torch.cat([self.mat, kv], dim=0)

class LIMeRouter(nn.Module):
    def __init__(self, config, layer_idx):
        super().__init__()
        bound = math.sqrt(
            3 / (layer_idx + 1) * config.num_kv_heads
        )
        weights = torch.empty(
            config.num_kv_heads,
            (layer_idx + 1) * config.num_kv_heads,
        ).uniform_(-bound, bound)
        weights[:, -config.num_kv_heads:] = torch.eye(
            config.num_kv_heads
        )
        self.weights = nn.Parameter(weights)

    def forward(self, kv_buffer):
        # kv_buffer shape = [(layer_idx + 1) * kv_h, 2 * b * t * hd]
        return self.weights.mm(kv_buffer)

class LIMeAttention(LlamaAttention):
    def __init__(self, config, layer_idx):
        super().__init__(config, layer_idx)
        if layer_idx > 0:
            self.lime_router = LIMeRouter(config, layer_idx)

    def forward(self, hidden_states, kv_buffer):
        query_states = self.q_proj(hidden_states).reshape(b, h, t, hd)
        key_states = self.k_proj(hidden_states).reshape(b, kv_h, t, hd)
        value_states = self.v_proj(hidden_states).reshape(b, kv_h, t, hd)
        kv_buffer.add_(key_states, value_states)
        if self.layer_idx > 0:
            key_states, value_states = self.lime_router(kv_buffer)
        attn_output = scaled_dot_product_attention(
            query_states, key_states, value_states
        )
        attn_output = self.o_proj(
            attn_output.transpose(1, 2).reshape(b, t, -1)
        )
        return attn_output, kv_buffer

class LIMeLayer(LlamaDecoderLayer):
    def __init__(self, config, layer_idx):
        super().__init__(config, layer_idx)
        self.self_attn = LIMeAttention(config, layer_idx)

    def forward(self, hidden_states, kv_buffer):
        residual = hidden_states
        hidden_states = self.input_layernorm(hidden_states)
        attn_out, kv_buffer = self.self_attn(hidden_states, kv_buffer)
```

```
63          hidden_states = residual + attn_out
64
65          residual = hidden_states
66          hidden_states = self.post_attention_layernorm(hidden_states)
67          hidden_states = self.mlp(hidden_states)
68          hidden_states = residual + hidden_states
69
70          return hidden_states, kv_buffer
71

72
73  class LIMeModel(LlamaModel):
74      def __init__(self, config):
75          super().__init__(config)
76          self.layers = [
77              LIMeLayer(config, i) for i in range(config.num_hidden_layers)
78          ]
79
80      def forward(self, input_ids):
81          hidden_states = self.embed_tokens(input_ids)
82          kv_buffer = KVBuffer()
83          for layer in self.layers:
84              hidden_states, kv_buffer = layer(hidden_states, kv_buffer)
85          return hidden_states
```

# I EFFICIENCY

| MHA | Model | # Parameters (B) | FLOPs (T) |
|---|---|---|---|
| GQA | LLaMa | 1.07607 | 2.7615 |
| | DenseFormer | 1.07607 (+0.00%) | 2.7622 (+0.02%) |
| | LIMe | 1.07608 (+0.00%) | 2.7638 (+0.08%) |
| | HC | 1.07640 (+0.03%) | 2.7701 (+0.31%) |
| Full | LLaMa | 1.17674 | 2.9679 |
| | DenseFormer | 1.17674 (+0.00%) | 2.9685 (+0.02%) |
| | LIMe | 1.17687 (+0.01%) | 3.0041 (+1.22%) |
| | HC | 1.17706 (+0.03%) | 2.9764 (+0.29%) |

Table 13: Model size (# parameters, in billions) and forward FLOPs for LIMe, Hyper-connections (HC), and DenseFormer relative to LLaMa under grouped-query attention (GQA) and full attention. We used `torch.jit.trace` to record all operations and estimated FLOPs via the `fvcore` library, based on tensor shapes and ATen operators. Total training FLOPs are approximated as $3\times$ forward FLOPs, accounting for both forward and backward passes (Anthony et al., 2023).

| MHA | RO | Model | Step Time (ms) | Train Peak Memory (GB) |
|---|---|---|---|---|
| GQA | + | LLaMa | 65.770 | 16.035 |
| | | LIMe | 66.533 (+1.16%) | 16.035 (+0.00%) |
| | | DenseFormer | 75.032 (+14.08%) | 16.812 (+4.85%) |
| | | HC | 81.003 (+23.16%) | 16.040 (+0.03%) |
| | − | LLaMa | 66.404 | 20.489 |
| | | LIMe | 67.449 (+1.57%) | 20.490 (+0.00%) |
| | | DenseFormer | 75.739 (+14.06%) | 21.646 (+5.65%) |
| | | HC | 83.265 (+25.39%) | 21.693 (+5.88%) |
| Full | + | LLaMa | 69.776 | 17.535 |
| | | LIMe | 77.093 (+10.49%) | 17.537 (+0.01%) |
| | | DenseFormer | 79.157 (+13.44%) | 18.348 (+4.64%) |
| | | HC | 84.990 (+21.80%) | 17.540 (+0.03%) |
| | − | LLaMa | 70.258 | 22.364 |
| | | LIMe | 77.607 (+10.46%) | 22.367 (+0.01%) |
| | | DenseFormer | 79.733 (+13.49%) | 23.566 (+5.37%) |
| | | HC | 86.314 (+22.85%) | 23.007 (+2.87%) |

Table 14: Per-step latency and peak GPU memory usage of LIMe, DenseFormer, and Hyper-connections (HC) in comparison to LLaMa under grouped-query attention (GQA) and full attention (Full), measured with PyTorch Inductor in default (−) and reduced-overhead (+) modes.

## J PIPELINE PARALLELISM

Under standard DDP training, LIMe does not incur any additional memory overhead—routing occurs via existing KV caches. Under pipeline parallelism (PP), the KV cache must be communicated across stages. However, we show that this can be efficiently implemented using asynchronous scheduling. Specifically, each pipeline stage:

- Computes its transformer layer output on already acquired micro-batch routed states.
- Routes KV buffers for later layers via non-blocking ops.

This dual-pipeline structure (forward pass + KV routing) allows communication and computation to be efficiently overlapped, minimizing idle time and avoiding runtime bottlenecks. Such scheduling strategies are well-established in modern pipeline parallelism frameworks, including DeepSpeed's PipeTransformer (He et al., 2021) and Megatron-LM (Shoeybi et al., 2019). While implementing a fully optimized schedule requires non-trivial engineering effort, we leave this for future work. To provide preliminary empirical evidence of scalability, we implemented pipeline parallelism for the 8B model using a straightforward 1F1B schedule across 8 stages (8 GPUs). In our measurements LIMe incurs only a **7.8%** training latency overhead (**1130** vs. **1048** ms/step), indicating that PP communication for routed KV can be efficiently hidden in practice.

## K LIME VISUALISATION

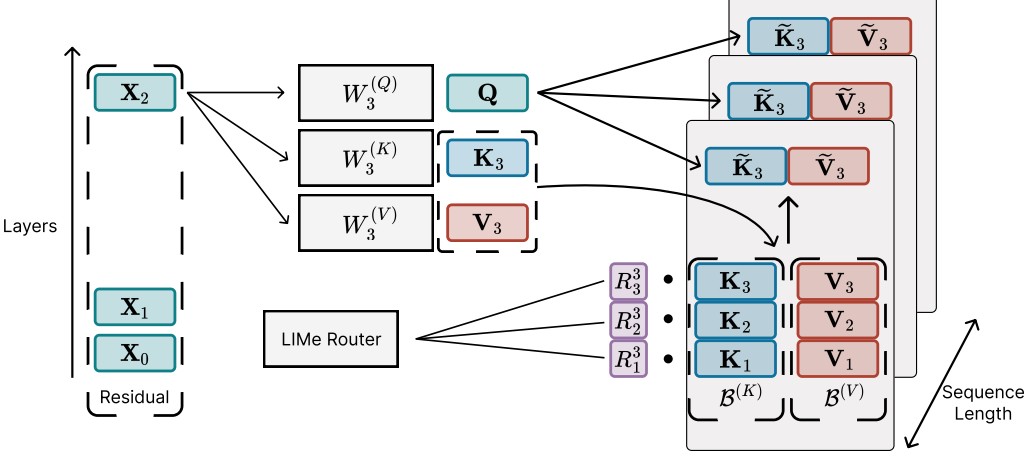

Figure 12: LIMe routing scheme.

## L LLM USAGE

We used LLMs for writing and text polishing.

