# OpenReview forum: "You Do Not Fully Utilize Transformer's Representation Capacity"
_ICLR.cc/2026/Conference — Submitted to ICLR 2026_

### Official Review · Reviewer_8E6q · 2025-10-30

**Soundness:** 3
**Presentation:** 2
**Contribution:** 3
**Rating:** 6
**Confidence:** 3

**Summary:**

This paper identifies "representation collapse" as a key weakness in standard Transformer decoders, where the reliance on a single residual stream from the immediately preceding layer forces the model to compress all prior information, leading to a loss of feature diversity in deeper layers. To address this, the authors propose LIMe, a lightweight architectural modification. LIMe allows each attention head at every layer to compute its KV representations by routing and mixing the KV buffers from all preceding layers, not just the current one. This is achieved by learning a per-head, per-layer routing matrix that weights the contributions of past layers.

**Strengths:**

- The primary strength of LIMe is its elegance and low overhead. By reusing existing KV buffers, it adds multi-layer information flow with almost no additional memory and a negligible computational cost (especially when GQA is used). This makes it a very practical and "drop-in" friendly modification.

-  The paper does an excellent job of clearly identifying a specific problem (representation collapse) and proposing a solution (LIMe) that directly targets it.

**Weaknesses:**

- The authors correctly identify in the limitations that the vanilla implementation of the router has an $\mathcal{O}(L^2)$ asymptotic complexity (where $L$ is the number of layers), as each layer's router must process keys from all $L-1$ previous layers. This is fine for the 16-layer models in the main paper, but it will become a significant computational bottleneck for scaling to very deep models (e.g., $L=100+$). The heuristic ablations in Appendix F (e.g., last-j or first-j) all show worse performance, suggesting a difficult trade-off between performance and scalability.
- The method's core idea, accessing all previous KV caches, creates a practical implementation challenge for large-scale training. In a standard pipeline parallel setup, this would require significant communication across pipeline stages (GPUs), as later layers would need to fetch KV caches from all earlier GPUs. The authors acknowledge this and their preliminary test shows a ~7.8% latency overhead. This practical hurdle might deter adoption for training SOTA-scale models, as it requires "non-trivial engineering effort" to optimize.

**Questions:**

Please refer to my weakness part.

---

> ### Author Response · Authors · 2025-11-21
>
> Thank you for the positive and detailed review.
>
> **W1.** We acknowledge that naïve LIMe has $O(L^2)$ routing in the number of layers ($L$), and that the phrase “negligible overhead” requires clarification and context. For our primary 16-layer experiments at the 1 B-parameter scale, LIMe with GQA adds approximately **+1.16 % latency** and **~0% memory overhead** relative to the baseline. We will revise our wording to explicitly specify that regime.
>
> | Model | Time / iter (ms) | Peak Mem (MB) | Perplexity |
> | --- | --- | --- | --- |
> | LLaMA | 70.21 | 2054.26 | 23.73 |
> | LIMe full | 80.85 (+15.2%) | 2062.38 (+0.4%) | 20.72 (–12.7%) |
> | LIMe, dilation=8 | 71.59 (+2.0%) | 2055.38 (+0.05%) | 21.61 (–8.9%) |
> | LIMe, dilation=16 | 71.57 (+1.9%) | 2054.88 (+0.03%) | 21.84 (–8.0%) |
> | LIMe, first-7 | 71.79 (+2.3%) | 2054.85 (+0.03%) | 20.55 (–13.4%) |
> | LIMe, first-15 | 72.69 (+3.5%) | 2055.76 (+0.07%) | 20.50 (–13.6%) |
>
> We also trained a **128-layer model** with several routing variants. The full LIMe (routing over *all* previous layers) improved perplexity from 23.73 (LLaMA) to 20.72 (–12.7 %) but increased per-step time by +15.2 %. However, lighter variants dramatically reduce both time and memory overhead, especially the **first-j routers** (where each layer reroutes to the first j layers plus itself), which scale **linearly in L** rather than quadratically, and indeed **outperform full LIMe**:
>
> - **first‑7:** 71.79 ms (+2.3 %), 20.55 perplexity (–13.4 %);
> - **first‑15:** 72.69 ms (+3.5 %), 20.50 perplexity (–13.6 %).
>
> Dilated routers (e.g., dilation = 8 or 16) have ~+2% time overhead and <0.1% memory overhead, while achieving 8–9% perplexity reductions. Figures 5 and 9 show that the router tends to concentrate most weight on **early layers**. In the 128-layer case, full LIMe forces the router to consider all L layers, so early-layer weights must adapt to noisier mid-layers, which slightly hurts performance. Restricting routing to the first j layers retains most of the informative early KVs and discards noisier mid-layers — this explains why first-j variants outperform both full and dilation variants.
>
> In summary, while naïve $O(L^2)$ routing becomes expensive at large depth, LIMe does **not** require it: linear-in-depth routing (e.g., first-j) retains essentially all the perplexity gains while incurring only ~2–3.5% latency and ~0% memory overhead even at 128 layers.

---

> ### Author Response · Authors · 2025-11-21
>
> **W2.** We agree with the reviewer that pipeline-parallel training introduces additional practical costs for LIMe’s KV routing, and we appreciate the prompt. Our prototype implementation uses a simple 1F1B schedule for an 8 B-parameter model over 8 pipeline stages and observes a **+7.8%** increase in step time (1130 vs 1048 ms). We acknowledge this as a **moderate but real limitation**, and we will clarify this as such in the revision. At the same time, we emphasise four supporting points:
>
> 1. **Relevance of pipeline parallelism.** Pipeline parallelism splits the model’s layers across multiple devices and transfers activations/states between stages; it is typically employed when a model *cannot* fit comfortably under pure data or tensor parallelism. For the 1B–32B regime, many widely-used deployments and training recipes rely solely on data or FSDP-style sharding (without pipeline parallelism), so in those settings, LIMe’s overhead is ~1–2% with GQA and zero in memory, making it effective for real production usage. 3D parallelism (data × tensor × pipeline) is typically required only at scales closer to hundreds-of-billions or trillion-parameter models [1, 2, 3].
> 2. **No additional persistent state beyond existing KVs.** Even under pipeline parallelism, LIMe does *not* introduce new large persistent buffers: we reuse the KV buffers that are already stored during standard attention in each pipeline stage. Our router simply reads from and blends these KVs, adding modest extra traffic that can be overlapped with computation.
> 3. **Compatibility with existing pipeline optimisations.** Scheduling algorithms like PipeDream [4] and its improved 1F1B schedule reduce pipeline bubbles by interleaving forward and backward passes across micro-batches; more advanced systems like PipeTransformer [5] employ adaptive layer freezing and elastic pipelining to deliver speedups without accuracy loss, while memory-balanced strategies like BPipe [2] enable asynchronous activation transfer to overlap communication with computation. LIMe is orthogonal to these developments and can benefit from overlapping its communication with computation in the same way.
> 4. **Reduced overhead of simpler routers.** As noted above, simpler routers (e.g., first-j) match or even outperform full LIMe in deeper models while reducing routing complexity from $O(L^2)$ to effectively $O(L)$. In a pipeline-parallel setup this means each stage only needs KVs from a small, fixed subset of earlier layers, so LIMe can be deployed with bounded, localised KV reuse and significantly lower cross-stage communication than the naïve “all previous layers” variant.
>
> We will therefore revise the paper to clarify that our “negligible overhead” claim is **targeted at non-pipelined models up to hundreds of billions of parameters**, present the +7.8% overhead as a **limitation specific to pipeline-parallel training of very large models**, and note that pipeline-aware scheduling (or simpler routers) can reduce it further. We thank the reviewer for prompting us to sharpen and contextualise these claims.
>
> We have expanded on all aspects you highlighted, and we kindly ask that you consider raising the score.
>
> **References**
>
> [1] Megatron-LM: Training Multi-Billion Parameter Language Models Using Model Parallelism, Shoeybi M. et al. (2019)
>
> [2] BPipe: Memory-Balanced Pipeline Parallelism for Training Large Language Models, Kim T. et al. (2023)
>
> [3] Advances of Pipeline Model Parallelism for Deep Learning Training: An Overview, Guan L. et al. (2024)
>
> [4] PipeDream: Generalized Pipeline Parallelism for DNN Training, Narayanan D. et al. (2019)
>
> [5] PipeTransformer: Automated Elastic Pipelining for Distributed Training of Large-scale Models, He C. et al. (2021)

---

### Official Review · Reviewer_nGJ6 · 2025-10-31

**Soundness:** 3
**Presentation:** 2
**Contribution:** 2
**Rating:** 4
**Confidence:** 4

**Summary:**

This paper starts from the observation that in standard transformer networks, there is a single residual stream, meaning that the representations from all the previous layers are compressed into a single hidden state. This single hidden state is then used as the input of the next layer. This can lead to *representation collapse*, which is the phenomenon where different tokens become undistinguishable. Hence, this paper propose a new mechanism to address this issue, called LIMe. The idea is that each layer can attend to the representations of *all* previous layers, instead of just the immediante previous one. In practice, this is done by modifying the way keys and values are computed. Instead of just using the keys and values computed from the input of the current layer, the keys and values from all previous layers are linearly combined, using trainable weights. Said otherwise, the keys and values of used in the attention of layer L are obtained by doing a linear combination of the keys and values of all the heads of the previous layers. The weight of this linear combination are fixed trainable parameters.

The proposed method is then empirically evaluated on different language modeling tasks. First, a LLaMa like model, with 1B parameters is trained on 50B tokens, and evaluated on downstream NLP tasks such as QNLI, WiC or ARC (easy/challenge). Here the experiments show that LIMe obtain better performance than the standard transformer architecture, as well as other approaches such as DenseFormer or HyperConnections. Then the model is compared to the standard transfomer on GSM8k or synthetic tasks such as arithmetic expression evaluation, again showing that LIMe performs better than the baseline. There are also ablations studying the *representation collapse* showing that LIMe is less prone to representation collapse than standard transformers.

**Strengths:**

I am a bit of the fence regarding this paper.

In terms of strengths, I believe that the proposed idea in the paper is simple and elegant. The paper is clearly written and easy to follow. The experimental evaluations are convincing.

**Weaknesses:**

My main concern with the paper is its relation to previous work, and especially its significance with respect to these.

First, I believe that the paper does not make a great job discussing the difference with previous work such as DenseFormer, or Value Residual Learning. More precisely, I think that the idea of combining the representations from multiple previous layers instead of just using the representation from the previous layer is not new. The contributions of the paper are thus mostly about details of how this idea is implemented in practice, and the paper could do a better job at discussing these. Moreover, I believe that the baseline considered in the paper (DenseFormer, HyperConnection) have multiple variant considered in the original papers, and the details of which one is used are missing. Finally, I am a bit surprised that the baseline (such as DenseFormer) does not seem to improve compared to the standard transformer, which goes against the claim of the original paper.

Another minor concern is the additional runtime required by the method, as it needs to read significantly more activations from memory compared to the standard transformer.

**Additional references**

*MUDDFormer: Breaking Residual Bottlenecks in Transformers via Multiway Dynamic Dense Connections.* Da Xiao, Qingye Meng, Shengping Li, Xingyuan Yuan. 2025.

*Value Residual Learning.* Zhanchao Zhou, Tianyi Wu, Zhiyun Jiang, Fares Obeid, Zhenzhong Lan. 2024

*LAUREL: Learned Augmented Residual Layer.* Gaurav Menghani, Ravi Kumar, Sanjiv Kumar. 2024

*DeepCrossAttention: Supercharging Transformer Residual Connections.* Mike Heddes, Adel Javanmard, Kyriakos Axiotis, Gang Fu, MohammadHossein Bateni, Vahab Mirrokni. 2025

**Questions:**

Which variant of DenseFormer and HyperConnection did you use?

Did you re-implement the baselines yourself or use existing code?

---

> ### Author Response · Authors · 2025-11-21
>
> Thank you for the constructive and thorough review.
>
> **W1.** We agree that our method LIMe lies in the broader family of transformer modifications that reuse earlier-layer representations (e.g., DenseFormer, HyperConnections, and Value Residual Learning). We have accordingly expanded the Related Work section to reflect these connections.
>
> Our contribution lies in **how** we perform this reuse, and **what effect** it has:
>
> 1. DenseFormer aggregates hidden states from previous layers and therefore must retain them in memory, whereas LIMe mixes per-head query-key/value (K/V) projections and reuses the existing K/V cache buffer, thus incurring no additional activation memory. Moreover, LIMe performs **per-head mixing across layers and heads**, and in Appendix F we now demonstrate that removing cross-head mixing degrades performance:
>
>
>     | Setup | Loss | Perplexity |
>     | --- | --- | --- |
>     | LLaMa | 2.80043 | 16.45 |
>     | LIMe default | 2.76889 | 15.94 (-3.1%) |
>     | LIMe no head mix | 2.83235 | 16.99 (+3.3%) |
> 2. The Value Residual Learning family (e.g., ResFormer/SVFormer) effectively reuses only the first layer’s values across depth; by contrast, LIMe generalises this to a learned per-head routing over K/Vs **from all previous layers** (or a subset such as first-j/dilated variants) and across heads. Empirically, we link this more flexible routing mechanism to reduced representation collapse and improved downstream performance.
>
> Regarding baseline details:
>
> - We train the DenseFormer variant with dilation = 1 and period = 1.
> - For HyperConnections, we adopt the Dynamic variant with expansion rate 4 (the best configuration reported in their original work).
>
> Regarding benchmark performance: although DenseFormer obtains a marginally better perplexity than LIMe and LLaMa (as in the original paper), this advantage does **not** translate into better downstream performance on language modeling benchmarks. We have added this table and discussion in Section 5 and Appendix A.
>
> | Model | Perplexity |
> | --- | --- |
> | LLaMA | 11.56 |
> | LIMe | 11.47 (–0.8%) |
> | DenseFormer | 11.36 (–1.7%) |
> | HyperConnections | 11.25 (–2.7%) |
>
> **W2.** We measured training latency on 1B-parameter models and report results in Table 14 (Appendix I). The results show that the overhead introduced by LIMe is negligible: on average, we observe a 1.16 % increase in per-iteration time, and near-zero extra memory is required.
>
> **Q1.** As noted above, for DenseFormer we used dilation = 1, period = 1; for HyperConnections we used the Dynamic variant with expansion rate 4. These specifications will be stated in the updated manuscript (Appendix A).
>
> **Q2.** We used the official implementation of DenseFormer from their GitHub repository. HyperConnections was implemented as described in the original paper (we used their released code). We clarify this in Section 4.2 of the revised submission.
>
> We have responded fully to each concern, and we respectfully request that you consider raising the score.

---

### Official Review · Reviewer_vzV7 · 2025-11-03

**Soundness:** 4
**Presentation:** 3
**Contribution:** 3
**Rating:** 4
**Confidence:** 4

**Summary:**

This paper proposes Layer-Integrated Memory (LIMe), which allows each attention head to access Key-Value representations from all previous layers through learned routing weights.

**Strengths:**

Comprehensive experimental design: The evaluation spans multiple dimensions: language modeling perplexity, mathematical reasoning on GSM8K, and synthetic tasks with controlled difficulty levels. The representation collapse analysis combines entropy measurements, linear separability tests, and grammatical probing to validate the core hypothesis from different angles. The routing weight analysis provides interpretability by revealing which layer representations the model prefers to access. This is the most lovely part of this paper.

**Weaknesses:**

1. Limited novelty over prior work. The core mechanism of using learned weights to aggregate multi-layer representations appears in Transparent Attention (Bapna et al., EMNLP 2018), which uses trainable softmax-normalized weights to combine encoder layer outputs in NMT decoder cross-attention. The mathematical formulation resembles that prior work, with the main difference being application to decoder-only self-attention. More recently, Hyper-Connections (Zhu et al., Sept 2024) addresses representation collapse through multi-stream connections with learned routing, sharing similar motivation. The paper does not clearly articulate what architectural insight LIMe provides beyond adapting these known techniques to decoder-only models with efficient KV buffer reuse.
2. Unclear computational cost analysis. The paper claims "negligible overhead" yet mentions O(L**2) routing complexity in limitations. For a 64-layer model, each layer must route over 64 previous layer KV pairs, but the paper does not provide memory bandwidth analysis for this case. The pipeline parallelism overhead of 7.8% contradicts the "negligible" claim for production scenarios.

**Questions:**

N/A

---

> ### Author Response · Authors · 2025-11-21
>
> Thank you for the careful and insightful review.
>
> **W1.** We acknowledge that LIMe is conceptually related to earlier work that aggregates multi-layer representations (e.g., Transparent Attention, DenseFormer, and Hyper‑Connections). Specifically, Transparent Attention uses trainable softmax-normalized weights to combine encoder layer outputs in cross-attention, while DenseFormer/Hyper-Connections aggregate multi-layer signals in other ways.
>
> LIMe shares the high-level idea of re-using past representations, but it differs in **three key ways**:
>
> 1. It uses a **simple per-head router** instead of multiple residual streams or heavy mixing modules, making it **lighter and faster** than methods such as Hyper-Connections.
> 2. It operates directly on **key–value projections** and **reuses the existing KV cache**, so at the 1 B parameter scale it adds essentially **no extra activation memory** compared to the baseline.
> 3. It performs **per-head mixing across both layers and heads**, which we show in ablation studies is important and provides a more expressive yet still efficient routing mechanism.
>
> In addition, our work contributes an extensive empirical analysis of **representation collapse** (measured via entropy, separability, probing), demonstrating that this specific KV-based routing not only belongs to that family of methods, but also **yields clear, well-characterised benefits in decoder-only language models**.
>
> **W2.** We agree that naïve LIMe has $O(L^2)$ routing in the number of layers ($L$), and that the phrase “negligible overhead” requires clarification and context. For our primary 16-layer experiments at the 1 B-parameter scale, LIMe with GQA adds approximately **+1.16 % latency** and **~0% memory overhead** relative to the baseline. We will revise our wording to explicitly specify that regime.
>
> | Model | Time / iter (ms) | Peak Mem (MB) | Perplexity |
> | --- | --- | --- | --- |
> | LLaMA | 70.21 | 2054.26 | 23.73 |
> | LIMe full | 80.85 (+15.2%) | 2062.38 (+0.4%) | 20.72 (–12.7%) |
> | LIMe, dilation=8 | 71.59 (+2.0%) | 2055.38 (+0.05%) | 21.61 (–8.9%) |
> | LIMe, dilation=16 | 71.57 (+1.9%) | 2054.88 (+0.03%) | 21.84 (–8.0%) |
> | LIMe, first-7 | 71.79 (+2.3%) | 2054.85 (+0.03%) | 20.55 (–13.4%) |
> | LIMe, first-15 | 72.69 (+3.5%) | 2055.76 (+0.07%) | 20.50 (–13.6%) |
>
> We also trained a **128-layer model** with several routing variants. The full LIMe (routing over *all* previous layers) improved perplexity from 23.73 (LLaMA) to 20.72 (–12.7 %) but increased per-step time by +15.2 %. However, lighter variants dramatically reduce both time and memory overhead, especially the **first-j routers** (where each layer reroutes to the first j layers plus itself), which scale **linearly in L** rather than quadratically, and indeed **outperform full LIMe**:
>
> - **first‑7:** 71.79 ms (+2.3 %), 20.55 perplexity (–13.4 %);
> - **first‑15:** 72.69 ms (+3.5 %), 20.50 perplexity (–13.6 %).
>
> Dilated routers (e.g., dilation = 8 or 16) have ~+2% time overhead and <0.1% memory overhead, while achieving 8–9% perplexity reductions. Figures 5 and 9 show that the router tends to concentrate most weight on **early layers**. In the 128-layer case, full LIMe forces the router to consider all L layers, so early-layer weights must adapt to noisier mid-layers, which slightly hurts performance. Restricting routing to the first j layers retains most of the informative early KVs and discards noisier mid-layers — this explains why first-j variants outperform both full and dilation variants.
>
> In summary, while naïve $O(L^2)$ routing becomes expensive at large depth, LIMe does **not** require it: linear-in-depth routing (e.g., first-j) retains essentially all the perplexity gains while incurring only ~2–3.5% latency and ~0% memory overhead even at 128 layers.

---

> ### Author Response · Authors · 2025-11-21
>
> We agree with the reviewer that pipeline-parallel training introduces additional practical costs for LIMe’s KV routing, and we appreciate the prompt. Our prototype implementation uses a simple 1F1B schedule for an 8 B-parameter model over 8 pipeline stages and observes a **+7.8%** increase in step time (1130 vs 1048 ms). We acknowledge this as a **moderate but real limitation**, and we will clarify this as such in the revision. At the same time, we emphasise four supporting points:
>
> 1. **Relevance of pipeline parallelism.** Pipeline parallelism splits the model’s layers across multiple devices and transfers activations/states between stages; it is typically employed when a model *cannot* fit comfortably under pure data or tensor parallelism. For the 1B–32B regime, many widely-used deployments and training recipes rely solely on data or FSDP-style sharding (without pipeline parallelism), so in those settings, LIMe’s overhead is ~1–2% with GQA and zero in memory, making it effective for real production usage. 3D parallelism (data × tensor × pipeline) is typically required only at scales closer to hundreds-of-billions or trillion-parameter models [1, 2, 3].
> 2. **No additional persistent state beyond existing KVs.** Even under pipeline parallelism, LIMe does *not* introduce new large persistent buffers: we reuse the KV buffers that are already stored during standard attention in each pipeline stage. Our router simply reads from and blends these KVs, adding modest extra traffic that can be overlapped with computation.
> 3. **Compatibility with existing pipeline optimisations.** Scheduling algorithms like PipeDream [4] and its improved 1F1B schedule reduce pipeline bubbles by interleaving forward and backward passes across micro-batches; more advanced systems like PipeTransformer [5] employ adaptive layer freezing and elastic pipelining to deliver speedups without accuracy loss, while memory-balanced strategies like BPipe [2] enable asynchronous activation transfer to overlap communication with computation. LIMe is orthogonal to these developments and can benefit from overlapping its communication with computation in the same way.
> 4. **Reduced overhead of simpler routers.** As noted above, simpler routers (e.g., first-j) match or even outperform full LIMe in deeper models while reducing routing complexity from $O(L^2)$ to effectively $O(L)$. In a pipeline-parallel setup this means each stage only needs KVs from a small, fixed subset of earlier layers, so LIMe can be deployed with bounded, localised KV reuse and significantly lower cross-stage communication than the naïve “all previous layers” variant.
>
> We will therefore revise the paper to clarify that our “negligible overhead” claim is **targeted at non-pipelined models up to hundreds of billions of parameters**, present the +7.8% overhead as a **limitation specific to pipeline-parallel training of very large models**, and note that pipeline-aware scheduling (or simpler routers) can reduce it further. We thank the reviewer for prompting us to sharpen and contextualise these claims.
>
> We have clarified all raised points in depth, and we would greatly appreciate it if you could consider increasing the score.
>
> **References**
>
> [1] Megatron-LM: Training Multi-Billion Parameter Language Models Using Model Parallelism, Shoeybi M. et al. (2019)
>
> [2] BPipe: Memory-Balanced Pipeline Parallelism for Training Large Language Models, Kim T. et al. (2023)
>
> [3] Advances of Pipeline Model Parallelism for Deep Learning Training: An Overview, Guan L. et al. (2024)
>
> [4] PipeDream: Generalized Pipeline Parallelism for DNN Training, Narayanan D. et al. (2019)
>
> [5] PipeTransformer: Automated Elastic Pipelining for Distributed Training of Large-scale Models, He C. et al. (2021)

---

### Official Review · Reviewer_Ax1G · 2025-11-04

**Soundness:** 3
**Presentation:** 2
**Contribution:** 3
**Rating:** 4
**Confidence:** 3

**Summary:**

The paper suggests adding a weighted average after the standard key projection. The average is taken over all the key representations of the current token in the current layer and head as well as the previous ones (over $i * h$ vectors in the $i$-th layer with a model having $h$ kv heads). The same is done for the values (but not the queries). The coefficient of weighted average is shared between keys and values. Results show improvement over baseline (as well as DenseFormer and HyperConnections) on downstream tasks. Particularly, there is a signficant boost in accuracy over Arithmetic Expression Task which is attributed to the ability to store more information needed for reasoning. Additionally the authors show that the representation remains linearly separable even in later layers which is not true about the baseline.

**Strengths:**

While the method shares similarity with existing methods such as DenseFormer, the correct placement of weighted averages is important and in addition to superior performance on the experiments, yields side-benefits such as the ability to re-use the KV cache. The authors report additional investigative results such as the analysis done on the learned router weights.

**Weaknesses:**

In Section 5.1, it would be very helpful to have the random baseline for each task. In particular, that results that are reported for several of tasks seem near-chance (e.g. WiC). There is also no confidence intervals reported which makes it very hard to determine the significance of the improvements. Overall this makes me question the efficacy of the method in general language modeling.

It is confusing to refer to LLaMA in Table 1. Based on my understanding, this is only a model with the same base architecture as LLaMA models where as a LLaMa baseline suggests the pre-trained models. I strongly suggest to make this clear since based on my understanding you are training everything from scratch.

I have asked additional questions below. Overall, I am uncertain about the intepretation of the provided results and whether they can currently clearly establish the effectiveness of the proposed method.

**Questions:**

1. When doing value classification (e.g. in Fig. 2b) is the rest of the model frozen?

2. Did you consider using a per-dimension (instead of per-head) weighted average? Was there any difference in performance? Alternatively, is it important to average across heads or is it enough to average over the same head across different layers?

3. DenseFormer does a similar mixing as the proposed method. Still, the results for DenseFormer are sometimes even worse than the baseline. Also, Denseformer paper reports reasonable improvements over the baseline. Why similar consistent improvements are not observed in these new set of experiments?

---

> ### Author Response · Authors · 2025-11-21
>
> Thank you for the thoughtful and detailed review.
>
> **W1.** In our revised manuscript, we now include **random-baseline results and standard errors** for all reported tasks in Tables 5, 6 and 7 in Appendix C. The updated tables are as follows:
>
> | Model | COPA (50) | MultiRC (50) | WiC (50) | QNLI (50) | WNLI (50) | Avg (50) |
> | --- | --- | --- | --- | --- | --- | --- |
> | LLaMA | **75.80 ± 1.92** | 43.24 ± 0.32 | 50.00 ± 0.89 | 49.49 ± 0.30 | 51.27 ± 2.66 | 53.96 |
> | DenseFormer | 74.00 ± 1.96 | 45.92 ± 0.32 | 49.69 ± 0.89 | 50.08 ± 0.30 | 52.11 ± 2.66 | 54.36 |
> | HC | 74.00 ± 1.96 | 54.34 ± 0.32 | 49.72 ± 0.89 | 49.43 ± 0.30 | **56.34 ± 2.64** | 56.77 |
> | **LIMe** | 75.20 ± 1.93 | **56.15 ± 0.32** | **50.44 ± 0.89** | **51.43 ± 0.30** | 56.06 ± 2.64 | **57.86** |
>
> | Model | ARC-E (25) | ARC-C (25) | HellaSwag (25) | OBQA (25) | Avg (25) |
> | --- | --- | --- | --- | --- | --- |
> | LLaMA | 70.45 ± 0.42 | 38.70 ± 0.64 | 52.55 ± 0.22 | 37.68 ± 0.97 | 49.85 |
> | DenseFormer | 70.60 ± 0.42 | 36.48 ± 0.63 | 41.46 ± 0.22 | 26.84 ± 0.89 | 43.85 |
> | HC | 71.15 ± 0.42 | 37.63 ± 0.63 | **54.04 ± 0.22** | **40.08 ± 0.98** | 50.73 |
> | **LIMe** | **71.15 ± 0.42** | **39.30 ± 0.64** | 52.85 ± 0.22 | 39.68 ± 0.98 | **50.75** |
>
> | Model | KV (50) | Induction (50) | IR (0.04) | CO (0.06) | Avg (25.03) |
> | --- | --- | --- | --- | --- | --- |
> | LLaMA | 45.94 ± 2.22 | 54.20 ± 2.69 | 12.94 ± 1.63 | 16.97 ± 0.38 | 32.51 |
> | DenseFormer | 50.30 ± 2.23 | 51.30 ± 2.69 | **15.76 ± 1.77** | **18.59 ± 0.39** | 33.99 |
> | HC | 51.68 ± 2.23 | 51.59 ± 2.69 | 15.29 ± 1.75 | 18.48 ± 0.39 | 34.26 |
> | **LIMe** | **55.64 ± 2.21** | **55.36 ± 2.68** | 14.82 ± 1.73 | 17.39 ± 0.38 | **35.80** |
>
> **W2.** Regarding the confusion around the term “LLaMA” in Table 1: Yes, we clarify in Section 5.1 that all models (including our baseline “LLaMA”) share the same base architecture (matching that of the publicly-known LLaMA models), but we **trained them from scratch** ourselves. We will revise the text to make this distinction clearer — the term “LLaMA” in the table refers to our reproduction of that architecture, not the off-the-shelf pre-trained model.

---

> ### Author Response · Authors · 2025-11-21
>
> **Q1.** Yes — in the value/hidden-state classification experiments, the underlying language model’s parameters are frozen. We collect the relevant representations and train only a simple logistic regression probe on top; the probe’s weights are the only trainable parameters.
>
> **Q2.** Yes — we performed an exploratory analysis of alternative parameterizations of the router architecture (now detailed in the Appendix F). Specifically, we tried:
>
> - **No head mixing**: each head in layer $\ell$ only mixes KV from **the** **same** **head index** across previous layers (router shape [H, L] instead of [H, L * H]).
> - **Per-dimension mixing**: each previous head is weighted by a `head_dim`-dimensional vector instead of a scalar (router shape [H, L * H * head_dim]).
>
> | Setup | Loss | Perplexity |
> | --- | --- | --- |
> | LLaMa | 2.80043 | 16.45 |
> | LIMe default | 2.76889 | 15.94 (-3.1%) |
> | LIMe no head mix | 2.83235 | 16.99 (+3.3%) |
> | LIMe per-dim mix | 2.77911 | 16.10 (-2.1%) |
>
> From this we draw two key takeaways:
>
> 1. **Cross-head mixing matters.** The “no head mix” variant eliminated the gains and in fact performed worse than baseline (+3.3% perplexity), implying that mixing across heads (not simply across layers within the same head) is important.
> 2. **Per-dimension routing is not necessary in our regime.** The per-dimension variant did improve over LLaMA (-2.1 %) but did *not* match the simpler scalar per-head router (-3.1 %) and introduced significantly more parameters (≈ head_dim × more) and cost. Hence, we opted for the scalar per-head router in the main method.
>
> **Q3.** The differences between our method LIMe and the compared method DenseFormer are twofold: memory usage and expressiveness.
>
> - In DenseFormer, the method must **store all previous layer outputs** in memory during inference.
> - In contrast, LIMe uses the existing Key-Value buffers and introduces a learned router that mixes attention heads from previous layers; thus, we enable richer cross-layer, cross-head mixing while **avoiding additional memory overhead**.
> - We also show that disabling head-mixing in LIMe leads to poorer performance, supporting the view that enhanced mixing is key:
>
>
>     | Setup | Loss | Perplexity |
>     | --- | --- | --- |
>     | LLaMa | 2.80043 | 16.45 |
>     | LIMe default | 2.76889 | 15.94 (-3.1%) |
>     | LIMe no head mix | 2.83235 | 16.99 (+3.3%) |
> - Moreover, the DenseFormer paper reports only perplexity for their trained models, whereas we report both perplexity and **real-benchmark downstream tasks**.
> - We used the official DenseFormer implementation, and the training results we obtained were: LLaMA 11.56, LIMe 11.47, DenseFormer 11.36 (perplexity). However, while DenseFormer shows lower perplexity in this controlled setting, it **did not** translate into improved performance on the downstream LM evaluation harness benchmarks we report (see Table 1). For transparency and reproducibility, the revised version now reports perplexity values for all baselines.
>
> We believe we have addressed all concerns comprehensively, and we kindly ask you to consider raising the score.

---

### Author Response · Authors · 2025-11-26

Dear Reviewers,

Thank you again for the time you have dedicated to evaluating our submission. We have uploaded a revised version of the paper that incorporates all clarifications and additional experiments discussed in our rebuttal. As the rebuttal period is nearing its end, we would kindly ask you to take another look at our responses and the updated paper, and to reconsider your scores if our clarifications adequately address your concerns.

We greatly appreciate your effort during this busy stage of the ICLR review process.

---

### Author Response · Authors · 2025-12-01

# General Response for the Area Chair

We thank the reviewers for their careful and constructive feedback. Reviewers broadly agree that LIMe is a simple, practical modification that reuses the KV cache with low overhead, improves perplexity and downstream reasoning performance, and is supported by a clear representation‑collapse analysis. The main concerns were about (1) relation to prior and concurrent work (vzV7, nGJ6), including DenseFormer’s behavior, (2) baselines and experimental clarity (Ax1G, nGJ6), and (3) efficiency and scalability (vzV7, 8E6q). Below, we summarize how the revised paper and discussion address each of these.

1. **Relation to prior / concurrent work and DenseFormer**

We now clarify more explicitly how LIMe fits into and differs from prior work that reuses earlier representations. The revised paper cites Value Residual Learning and LAUREL alongside DenseFormer and HyperConnections, and it marks MUDDFormer and DeepCrossAttention as concurrent work targeting residual bottlenecks or strengthened residual paths. In our responses, we explain that LIMe is distinct in that it

- operates directly at the KV level,
- reuses the existing KV cache with no extra activation memory at our main scale, and
- uses a per‑head router that mixes across both layers and heads.

New ablations (no head mixing, per‑dimension routing, etc.) show that this KV‑level, per‑head cross‑layer/head routing is necessary for the observed gains: removing cross‑head mixing or replacing the scalar router with a heavier per‑dimension variant consistently harms performance.

To address the concern that DenseFormer underperforms our baseline, we provide perplexity numbers for all the compared methods in discussion with the reviewer. DenseFormer and HyperConnections do achieve lower perplexity than our LLaMA baseline (consistent with their original claims), but this does not translate into better downstream accuracy, whereas LIMe improves over both. This clarifies that DenseFormer is not “broken” in our implementation; instead, LIMe offers a different trade‑off that better aligns perplexity and downstream performance.

2. **Baselines and experimental clarity**

We added random baselines and standard errors for all LM‑Harness benchmarks and clarify that all models (including “LLaMA”) are trained from scratch with the same architecture. For baselines, we now explicitly state that we use DenseFormer (dilation = 1, period = 1) and HyperConnections Dynamic (expansion rate 4), following strong variants from the original papers. DenseFormer is implemented using the official author code from the public GitHub repository, and the HyperConnections Dynamic implementation is taken directly from the code listing in the original paper. For probing experiments, we clarify that the language model is frozen and only a logistic‑regression probe is trained on top of the collected representations.

3. **Efficiency and scalability**

For our main 16‑layer, 1B models without pipeline parallelism, LIMe with GQA incurs about +1.16% latency and ≈0% extra activation memory; the “negligible overhead” claim is now explicitly scoped to this regime. To study depth scaling, we report 128‑layer results and show that structured routers (e.g., first‑j, dilated) reduce routing complexity to effectively $O(L)$ while retaining almost all perplexity gains and, in some cases, outperforming full $O(L^2)$ routing.

For pipeline parallelism, we explicitly report a +7.8% step‑time overhead for an 8B model with a simple 1F1B schedule and present this as a moderate but real limitation. We also explain how LIMe’s reuse of existing KV buffers and structured routing can localize communication and reduce this cost, and emphasize that many 1B–32B deployments rely on data‑ or FSDP‑style parallelism where the overhead remains ≈1–2%.

Taken together, these revisions directly address the reviewers’ concerns about novelty, baselines, and efficiency. We believe the revised paper now provides a clear, well‑supported contribution on alleviating representation collapse via KV‑level routing in decoder‑only transformers, and we respectfully submit that it merits acceptance.

---

### Meta-Review · Area_Chair_vDys · 2026-01-05

**Summary:**

Most reviewers appreciate the elegance of the proposed idea and the improved efficiency compared to previous works by reusing the KV cache to avoid the memory overhead. Multiple reviewers also praise the comprehensive evaluation in downstream tasks. The analysis of representation collapse is also helpful.

Reviewers share the following concerns:
1. Limited novelty. The idea of using hidden states of previous layers has been studied in multiple prior works. The contribution of this work is in the implementation of this idea but the authors failed to clarify the difference from prior works.
2. Benchmark performance. One reviewer is concerned of the advantage of LIMe over existing works in the benchmarks in lack of random baselines and standard errors
3. Efficiency and scaling. Multiple reviewers are concerned of the runtime overhead and practicality of the method in large-scale training especially with pipeline parallelism
4. Clarification questions about the implementation of baselines

**Reviewer Concerns:**

Addressed: concern 4. The rebuttal explains the discrepancy of DenseFormer's performance in this submission and in the original paper due to the use of different (training vs downstream) metrics.

Partially addressed: concern 1 and 3.
Concern 1. The rebuttal expands the discussion of the implementation difference between LIMe and existing works. It clarifies the benefits in reducing memory by reusing the KV cache, but the general concern of the similarity of the high level idea to existing works remain.
Concern 3. The rebuttal explains that the runtime overhead is negligible when using first-j routers with more details. However, it also acknowledges the "moderate but real limitation" of the runtime cost with pipeline parallelism which corroborates the concern for large-scale training.

Failed to addressed: concern 2. The rebuttal provides standard errors of the experimental results. LIMe is better than the baseilne, HC, with statistical significance in 5 out of 14 tasks, and worse in 2 tasks. The new results confirms the reviewer's concern on its empirical strength.

**Reviewer Scores:**

Ax1G: initial rating 4. The main concern is about the empirical performance. The rebuttal will not improve the reviewer assessment, if not lowering that.

vzV7, nGJ6: initial rating 4. The main concerns from both reviewers are the limited novelty and the computational cost analysis. As explained are the previous section, both concerns are only partially addressed. The rebuttal will not change their score.

8E6q: initial rating 6. The reviewer's main concern is the L^2 complexity and practical implementation challenge for large-scale training. The rebuttal explains the benefit of "first-j routers" with a linear complexity but at the same time acknowledges the practical challenge with pipeline parallelism. The rebuttal will not change the score.

---

### Decision · Program_Chairs · 2026-01-26

Reject